# SaVeNet: A Scalable Vector Network for Enhanced Molecular Representation Learning

**Sarp Aykent**
Auburn University
`sarp@auburn.edu`

**Tian Xia**
Auburn University
`tian@auburn.edu`

## Abstract

Geometric representation learning of molecules is challenging yet essential for applications in multiple domains. Despite the impressive breakthroughs made by geometric deep learning in various molecular representation learning tasks, effectively capturing complicated geometric features across spatial dimensions is still underexplored due to the significant difficulties in modeling efficient geometric representations and learning the inherent correlation in 3D structural modeling. These include computational inefficiency, underutilization of vectorial embeddings, and limited generalizability to integrate various geometric properties. To address the raised concerns, we introduce an efficient and effective framework, **Sca**lable **Ve**ctor Network (SAVENET), designed to accommodate a range of geometric requirements without depending on costly embeddings. In addition, the proposed framework scales effectively with introduced direction noise. Theoretically, we analyze the desired properties (i.e., invariance and equivariant) and framework efficiency of the SAVENET. Empirically, we conduct a comprehensive series of experiments to evaluate the efficiency and expressiveness of the proposed model, which achieves state-of-the-art performance across various tasks within molecular representation learning.

## 1 Introduction

The field of geometric deep learning (GDL) has seen a rapid expansion in recent years, thanks to the successful application of Graph Neural Networks (GNNs) for modeling graph structures [4]. The ability to learn complex geometric representations has driven significant breakthroughs across various disciplines and has proven to be particularly beneficial in diverse areas such as social science [24], physics [14], and neuroscience [6]. Within the realm of molecular representation learning, the use of message-passing-based GNNs has demonstrated remarkable outcomes, especially in understanding the 3D structures of molecules.

Although geometric deep learning has recently seen promising results in various applications in molecular representation learning, its full potential in the field is still untapped. To begin with, a notable research gap persists within this field: the trade-off between expressiveness and efficiency in GNNs. Many studies often resort to the use of multi-hop neighbors [9, 17, 21] or complex embeddings [19] in their pursuit to augment the expressiveness of message-passing based GNNs [9, 29, 33]. While these approaches can enhance the expressiveness and predictive power of the model, they invariably compromise scalability and efficiency due to their computationally intensive nature. As these networks grow more expressive, they demand more computational resources, impeding their ability to scale to larger or more complex datasets. Addressing the challenge of efficiently handling molecular structures to sufficiently capture the complete, multi-level structural information during learning without compromising the computational efficiency of the network is of critical importance. Our approach circumvents these issues by avoiding using multi-hop GNNs and

37th Conference on Neural Information Processing Systems (NeurIPS 2023).

expensive embeddings. Novel approaches for initializing and processing vector-typed embeddings have been designed to work with vector representations with the goal of achieving a balance between maintaining numerical stability, facilitating faster convergence, and enhancing the model's ability to generalize to new datasets [25, 33]. Utilizing one-hop vector embeddings work with our proposed SAVENET significantly enhances the network's efficiency, demonstrating their effective integration in this context. Figure 1 provides a visual illustration of the advantages of our proposed SAVENET , which compares our method with state-of-the-art methods on the QM9 dataset. The values on the *x*-axis show the latency of the model, and the values on the *y*-axis show the standard mean absolute error (std. MAE) of the models' performance. Because lower is better for both latency and std. MAE, the best model in both efficiency and expressiveness is the one that is closest to the (0,0) point. In terms of performance, our base model, denoted as SAVENET -B, surpasses the existing state-of-the-art method Equiformer [19], while concurrently enhancing training duration efficiency by a factor of 14.6. This substantial improvement underscores the efficacy of our model in achieving superior results with increased efficiency.

Furthermore, although several prior works [2, 3, 12, 18, 19, 25, 27, 29, 30] proposed equivariant neural networks that handle scalar and vector representations together for molecular representation learning, one primary problem with these existing works is that vector-valued representations show limited performance or scalability and cannot equally contribute towards performance improvements in the network [12, 18, 29]. It remains challenging to scale the models to improve the expressiveness of the model further. We propose an efficient framework to tackle this challenge. Our model's scalability is exemplified by its capacity to stack $N$ layers, effectively serving as an encoder for learning the latent geometric representations, thereby augmenting the model's

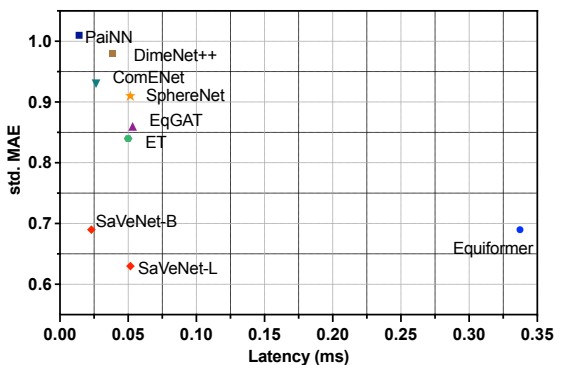

Figure 1: Comparison of our proposed SAVENET and the state-of-the art methods: latency vs. std. MAE on QM9 dataset.

expressiveness. These superior scaling capabilities enhance the model's expressiveness and effectively distinguish our work from existing research, which often struggles to scale efficiently [9, 19]. The scalability of our approach is not merely theoretical but is empirically demonstrated in Section 5.1. Figure 1 illustrates one example of the scaled larger version of our model: SAVENET -L, which exhibits enhanced capability in contrast to our base model and existing baselines. This advancement substantiates the prospective capacity of our model to scale effectively, thereby indicating potential for superior performance in future implementations.

Our paper's main contributions are: (i) Proposed an efficient and effective message-passing framework coupled with several decoder mechanisms for leveraging the geometric representations within the 3D molecular graphs. (ii) Novel approaches for initializing and processing vector-typed embeddings have been designed to work with vector representations to ensure numerical stability, speed up model convergence, and improve the model's ability to adapt to new datasets. (iii) A thorough evaluation of the expressiveness, efficiency, and scalability of our models compared with existing benchmarks, as well as detailed ablation studies, underscore the advantages of our proposed SAVENET .

## 2 Related work

**Equivariant Graph Neural Networks** The properties of graph representations can change with transformations such as translation, rotation, and permutation. By design, GNNs inherently permutation equivalence [15]. While it is possible to introduce diverse transformations to the network through data augmentation, this approach can lead to computational inefficiencies.

The concepts of invariance and equivariance have been long acknowledged as critical to the success of various tasks [2, 13, 35, 36]. Recent research in molecular representation learning has predominantly addressed this challenge with equivariant network architectures. Within the realm of molecules,

numerous equivariant models have been proposed for tasks such as chemical property prediction [16], protein structure prediction [13], and energy prediction [17].

**Molecular Property Prediction** Over recent years, there have been considerable advancements in molecular property prediction [9, 16, 19, 28]. These developments have been primarily driven by the adoption of graph models for molecular representation, coupled with the application of kernel methods for structural learning. Such kernels operate based on geometric features, such as interatomic distances, which are encoded using radial basis or Bessel basis functions. Several other studies employ spherical embeddings to model message-passing [3, 9, 17, 19, 21, 33]. Dimenet++ [9], GemNet [17] and SphereNet [21] depend on a multi-hop message-passing to model angular properties, which introduces computational complexity that curtails scalability with respect to the model size of these approaches. These advanced representations have resulted in more precise property predictions, including but not limited to dipole moment, HOMO-LUMO gap, energy, and force. However, the precision of these predictions is accompanied by increased computational complexity due to their dependence on quadratic or even cubic operations. Recent works have strived to address this issue, proposing innovative techniques that improve computational efficiency with minimal or no detriment to prediction accuracy [21, 33]. These novel techniques continue to operate on higher-order features during feature generation, underscoring the ongoing challenge of balancing computational efficiency with predictive precision. Both ET [30] and EQGAT [18] incorporate attention mechanisms into vector representations. However, the complex interactions in these works often limit the scalability and efficiency of the networks. More recently, networks such as SEGNN [3] and Equiformer [19] have utilized Clebsch-Gordan tensor products to achieve high performance. However, these tensor products are empirically found to be significantly slower than the previously discussed methods, posing a challenge when applying such methods to large, real-world datasets.

Our model utilizes 1-hop geometric representations, offering a balance of expressivity and computational efficiency. This design choice circumvents the need for complex and potentially lossy conversions often associated with higher-order representations. By harnessing the rich information in 1-hop representations, we maintain data integrity and enhance model performance without substantial computational overhead.

## 3 Preliminaries

**Notations.** We formulate the atomic structure graph representation, where each input molecule is represented as a graph $\mathcal{G} = (\mathcal{V}, \mathcal{E})$. $\mathcal{V}$ is the set of $N$ nodes in the graph, each node representing an atom in the molecule, and $\mathcal{E} \subseteq \mathcal{V} \times \mathcal{V}$ is the set of edges that connects the nodes. The average degree of a node, denoted by $k$, represents the average number of edges per node. The geometric information of a node for each $v_i \in \mathcal{V}$ is represented by the coordinates $c_i = (x_i, y_i, z_i)$ in the Cartesian coordinate system. The scalar representations $s$ generally include properties like relative distance $r$, while the vector representations $V$ include direction vectors $\vec{d}$. We use $\odot$ to denote the element-wise (Hadamard) product and $|| \cdot ||_2$ to denote the row-wise $L^2$ norm. The cross-product between two vectors is denoted as $\times$.

**Invariance and Equivariance.** A function $f : X \rightarrow Y$ is said to be *equivariant* to a transformation if applying the transformation to the input of the function is the same as applying the transformation to the output of the function. Formally, for a transformation $T$, a function $f$ is equivariant if for all $x \in X$: $T(f(x)) = f(T(x))$. This means that the function preserves the structure of the transformation. On the other hand, a function is *invariant* to a transformation if its output does not change when the transformation is applied to its input. Formally, for a transformation $T$, a function $f$ is invariant if for all $x \in X$: $f(T(x)) = f(x)$. This implies that the function is invariant to the transformation; the function's output remains unchanged regardless of the applied transformation.

**Euclidian Transformations.** In this paper, we frequently use the special orthogonal group $SO(3)$, comprising all 3x3 rotation matrices $R$ that satisfy $R^T R = I$ and $\det(R) = 1$, and the special Euclidean group $SE(3)$, consisting of all $4 \times 4$ special Euclidean transformation matrices $T$ that can be written as $T = \begin{bmatrix} R & t \\ 0 & 1 \end{bmatrix}$, where $R \in SO(3)$, $t \in \mathbb{R}^3$ is a translation vector, and $0$ is a row vector of zeros. These matrices satisfy $T^{-1}T = I$ and $\det(T) = 1$. $SE(3)$ forms a manifold, the product of $SO(3)$ and $\mathbb{R}^3$, allowing operations combining rotations and translations in 3D space.

## 4  Scalable Vector Network

Our goal is to design a message-passing neural network that is aware of the geometric characteristics of the molecular graph, with consideration of expressiveness and efficiency, and preserve these characteristics beneficial for downstream tasks. We first introduce the representations used in SAVENET and how we handle these geometric representations. Then we describe the overall message-passing framework to model the geometric properties in the graph and show how to integrate our proposed framework into the representation learning process on downstream tasks.

### 4.1  Efficient Representations for 3D graphs

The initial representations fed into the network are instrumental in achieving complete geometric representations [21, 33]. It is crucial to efficiently represent geometric graphs without the loss of their properties. Our framework operates by employing embeddings from one-hop neighbors exclusively. Invariant properties are encoded as scalar values, which enable the network to maintain invariance without resorting to computationally expensive operations. For equivariant properties, we extract the inherent directional information from the input graphs and encode these features into vector representations. The directional information between nodes $i$ and $j$ is represented with $\vec{\beta}_{ij}$ and defined as follows:

$$\vec{\beta}_{ij} = \left\{ \vec{d}_{ij} = \frac{(c_i - c_j)}{r_{ij}}, \ \vec{t}_{ij} = (c_i - \bar{c}) \times (c_j - \bar{c}), \ \vec{o}_{ij} = \vec{d}_{ij} \times \vec{t}_{ij} \right\} \in \mathbb{R}^{3 \times 3} \tag{1}$$

Here, $\bar{c}$ represents the average of $c$, and $\times$ denotes the cross-product operation. With the geometric representation defined, the efficacy of our scalar and vector representations is further demonstrated in the following lemma:

**Lemma 1.** *Consider a known geometric graph with at least one node, and assume that each node in this graph has at least one connection. Let a new node $j$ be added to this graph, stipulating that $j$ is connected to at least one existing node. Then, the position $p_j$ of node $j$ can be determined in constant time using the $(p_0, r_{ij}, \vec{d}_{ij})$ properties.*

The proof for Lemma 1 is provided in the Appendix A. Lemma 1 can be further generalized, requiring no positional information by translating the $p_0$ reference point to the origin using a $t_o$ translation vector. In light of this, we propose the following theorem, which demonstrates the SAVENET's ability to reconstruct the rotation-equivariant geometric structure of an input graph:

**Theorem 1.** *Assuming the input graph is strongly connected, the input space of SAVENET , defined as $(r, d)$, is capable of reconstructing the rotation-equivariant geometric structure of the input graph. The reconstructed graph maintains the relative spatial configuration of nodes and edges in the input graph, the translation transformation is required for exact alignment with the original graph.*

The proof for Theorem 1 is provided in Appendix B. The vector representations are an effective way to model equivariant networks and are widely used networks [11, 12, 25, 29] on different challenging tasks. With the proof of Lemma 1 and Theorem 1, we present a provably efficient and lossless geometric encoding for the 3D graphs without reliance on the multi-hop neighborhoods. Despite the vector representation offering effective encodings, several unresolved challenges hinder its successful application towards enhancing the model's expressiveness and scalability. In response to this, we propose the incorporation of direction noise initialization and a vector activation class within our SAVENET framework, specifically designed to address this issue.

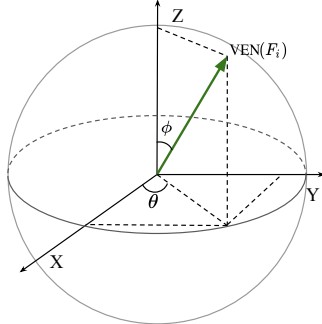

Figure 2: Direction noise.

**Direction Noise.** Vector representations are typically initialized using dataset-specific features [12, 25], such as direction vectors between sequential neighbors in a protein's amino acid sequence. However, in the absence of such data, existing works [18, 29] often default to initializing these vectors as null, allowing the network to learn these vector representations intrinsically. Such initialization can unnecessarily consume computational resources, particularly in the network's initial layers. In this work, as illustrated in Figure 2, we propose a novel strategy of

introducing directional noise utilizing the spherical coordinate system as an initialization measure for the vector representations generated based on the node type. This approach is designed to mitigate numerical instability issues and empower the network to explore the latent space we propose through direction noises. We follow the following convention to generate a direction vector for a given node type $F$:

$$\text{VEN}(F_i) = \left\{ \gamma(\sin\theta_i \cos\phi_i), \gamma(\sin(\theta_i)\sin(\phi_i)), \gamma(\cos(\theta_i)) \mid \theta_i \in \mathbb{R}, \phi_i \in \mathbb{R} \right\} \tag{2}$$

where $\theta_i$ and $\phi_i$ are learnable parameters for node type $i$ and $\gamma$ is a control parameter governing the magnitude of the direction embeddings. This approach effectively generates direction vectors using the spherical coordinate system. The value of $\gamma$ is associated with the progress of the training, similar to learning rate schedulers. The value of $\gamma$ is adjusted throughout training, following a decay schedule that ensures it reaches zero before training concludes. This methodology offers an effective means for creating direction vectors using a spherical coordinate system while allowing for the adaptive control of the direction embeddings' magnitude throughout training.

**Vector Activations.** Applying similar activation functions used in scalars on vector representations, where the function operates on each element of the vector independently, can break the model's equivariance. This is because these functions are not designed to handle the geometric transformations that the vectors might undergo. Consequently, SAVENET introduces a vector activation function in which the activation is computed based on the vector's $L^2$ norm. For vector representation $V$, the vector activation function is written as follows:

$$\text{VA}(V) = \left( V \odot \sigma_v(||V|| + b) \mid V \in \mathbb{R}^{C \times 3}, b \in \mathbb{R}^C \right) \tag{3}$$

where $\sigma_v$ is a scalar activation function. To further regulate the activation, the vector activation function incorporates a bias term $b$, where $b$ is the same shape as the representation dimension $C$. Our experimentation found that vector activation functions significantly enhance the numerical stability during training, particularly with larger-scale models.

## 4.2  SAVENET for 3D Graphs

This section describes the overall architecture of SAVENET in detail. The overall SAVENET framework is a SO(3)-equivariant message-passing neural network incorporating the vector initialization and activation for scalable molecular representation learning. It is based on an encoder-decoder structure comprising multiple stacked encoder and decoder layers. The encoder layers learn the latent representation of the geometric features, and the decoder layers decode the learned latent representations for downstream tasks on either invariant or equivariant targets.

**Encoder.** The encoder in SAVENET follows the message-passing paradigm and incorporates the above-defined geometric representations. The main two components of the encoder are inter-atomic interactions followed by atom-wise blocks.

*Vector Updates.*  Vector updates are employed to compute channel-wise interactions in vector representations. By conducting identical operations across spatial dimensions, vector updates ensure that the equivariance property is maintained. More specifically, given a vector representation $V \in \mathbb{R}^{C \times 3}$, the vector update applies a learnable weight matrix $\mathbf{W} \in \mathbb{R}^{C \times C'}$, where $C$ and $C'$ denote the dimensions of the input and output representations, respectively. Therefore, the output vector $V'$ is computed as $V' = \mathbf{W}V$.

*Inter-atomic interactions.*  Our model encapsulates inter-atomic interactions by integrating both equivariant direction vectors and invariant distance filters, which are encoded using radial basis functions. We have designed the scalar interaction path to interface with the vector interactions in a manner that preserves the equivariance property, achieved by appropriately scaling the magnitudes of the vectors. The overall formulation of the inter-atomic interactions can be expressed as follows:

$$\begin{aligned} IA(s, V, r, \vec{\beta}) &= (s_i, V_i) + \sum\nolimits_{j \in \mathcal{N}_i} e(s_j, V_j, r_{ij}, \vec{\beta}_{ij}) = (s'_i, V'_i) \\ s'_i &= s_i + \sum\nolimits_{j \in \mathcal{N}_i} e_s(s_j, V_j, r_{ij}, \vec{\beta}_{ij}) \\ V'_i &= V_i + \sum\nolimits_{j \in \mathcal{N}_i} e_v(s_j, V_j, r_{ij}, \vec{\beta}_{ij}) \end{aligned} \tag{4}$$

In this equation, $\mathcal{N}_i$ represents the set of neighbors of node $i$, and $e$ is the interaction function, $\vec{\beta}$ denotes the equivariant vectors. To accommodate a variety of equivariant properties, we fuse these vectors with vector updates VA. The interaction function is formalized as follows:

$$
\begin{aligned}
e_s(s_j, V_j, r_{ij}, \vec{\beta}_{ij}) &= \phi_s(s_j) \odot \eta_s(r_{ij}) \\
e_v(s_j, V_j, r_{ij}, \vec{\beta}_{ij}) &= \phi_b(\vec{\beta}_{ij}) \odot \phi_d(s_j) \odot \eta_d(r_{ij}) + V_j \odot \phi_v(s_j) \odot \eta_r(r_{ij})
\end{aligned}
\tag{5}
$$

where $\phi$ denotes sequentially stacked perceptron layers and $\eta$ denotes distance encoding. Specifically, $\phi_b$ serves as a vector update layer as it operates on vector inputs. The distance encoding is calculated using the formula $\eta(r_{ij}) = \phi_r(\chi(r_{ij}) * \omega(r_{ij}))$, where $\chi(\cdot)$ denotes a *basis* function and $\omega(\cdot)$ stands for a cutoff function.

*Atom-wise blocks.* Following the execution of the interaction block, the workflow proceeds to the atom-wise blocks, which are responsible for computing interactions between invariant and equivariant representations, as well as performing channel-wise updates. The formal expression of this process is defined as:

$$
\text{AW}(s, V) = \left\{ \left( \phi_m(\|\phi_{vu}(V)\| \cup s) \right), \left( \text{VA}(\phi_{vu}(V) \odot \phi_v(\|\phi_{vu}(V)\| \cup s)) \right) \right\}
\tag{6}
$$

where $\phi_{\{m,v\}}$ denotes multi-layer perceptron (MLP) and $\phi_{vu}$ denotes vector updates. Here, the vector representations interact with scalar representations via the $L^2$ norm of updated vector representations. Moreover, the scalar representations scale the vector representations after applying channel-wise interactions $\phi_v$. This process presents bi-directional communication between different representations, thus enhancing the model's ability to learn and extract meaningful information from data.

**Decoder.** The framework utilizes decoder variations to accommodate various geometric requirements of a given task. By taking the latent representation pair $(s_e, V_e)$ from the output of the encoder, the decoder can effectively leverage these representations to maximize task performance. For *invariant targets* where the tasks do not depend on global rotations, transformations, and permutation, the decoder takes encoded $s_e$ representations and processes them with MLP layers. For *equivariant targets* where tasks are sensitive to global rotations, the decoder takes the scalar and vector representations and processes them with stacked AW blocks. The final vector representations are scaled with scalar representations to generate the predictions. If the task is graph level, final representations are aggregated with global sum pooling.

**Equivariance and Invariance.** Addressing the strategic initialization of the model and the design of operations that uphold invariant and equivariant properties is equally important. SAVENET uphold such properties, which can be stated more formally as follows:

**Theorem 2.** *The equivariant representation of* SAVENET *is equivariant to any given rotation matrix in $R \in SO(3)$. The invariant representation of* SAVENET *is invariant to any given transformation matrix in SE(3).*

The formal proof of Theorem 2 is provided in Appendix C, demonstrating the validity of these assertions and the robustness of the model under various transformations.

## 5 Experiments

This section evaluates our proposed SAVENET on three tasks in geometric representation learning over both synthetic and real-world datasets. We implement the model with PyTorch [26]. All experiments are conducted on an NVIDIA 3090 GPU with 24 GB memory. All models use the AdamW optimizer [22] for the optimization. Detailed descriptions of the datasets are compared in Table 1, which includes the average number of nodes, the average number of edges, and the total number

Table 1: Dataset details. The number of edges are computed with radius graph where d(Å) = 5.0

| Datasets | QM9 | N-Body | Molecule3D |
|---|---|---|---|
| Avg. # of Nodes | 18.02 | 20 | 29.11 |
| Max # of Nodes | 29 | 20 | 137 |
| Avg. # of Edges | 280.72 | 380 | 553.00 |
| # of Graphs | 130,831 | 7,000 | 3,899,647 |
| # of Tasks | 12 | 4 | 5 |
| Splits | 84:8:8 | 42:29:29 | 6:2:2 |
| Invariant | ✓ | ✗ | ✓ |
| Equivariant | ✗ | ✓ | ✓ |

Table 2: Performance comparisons on QM9. † denotes using different data partitions.

| Task | $\alpha$ | $\Delta\varepsilon$ | $\varepsilon_{\text{HOMO}}$ | $\varepsilon_{\text{LUMO}}$ | $\mu$ | $C_\nu$ | $G$ | $H$ | $R^2$ | $U$ | $U_0$ | ZPVE | std. | log |
|------|------|------|------|------|------|------|------|------|------|------|------|------|------|------|
| Units | $a_0^3$ | meV | meV | meV | D | $\frac{\text{cal}}{\text{mol K}}$ | meV | meV | $a_0^2$ | meV | meV | meV | % | - |
| NMP† | .092 | 69 | 43 | 38 | .030 | .040 | 19 | 17 | .180 | 20 | 20 | 1.50 | 1.78 | -5.08 |
| SchNet | .235 | 63 | 41 | 34 | .0330 | .0330 | 14 | 14 | .073 | 19 | 14 | 1.70 | 1.76 | -5.17 |
| Cormorant† | .085 | 61 | 34 | 38 | .038 | .026 | 20 | 21 | .961 | 21 | 22 | 2.03 | 2.14 | -4.75 |
| LieConv† | .084 | 49 | 30 | 25 | .032 | .038 | 22 | 24 | .800 | 19 | 19 | 2.28 | 1.35 | -4.99 |
| PhysNet | .061 | 42.5 | 32.9 | 24.7 | .0529 | .0280 | 9.4 | 8.42 | .765 | 8.34 | 8.15 | 1.39 | 1.37 | -5.35 |
| DimeNet | .047 | 34.8 | 27.8 | 19.7 | .029 | .0249 | 8.98 | 8.11 | .331 | 7.89 | 8.02 | 1.29 | 1.05 | -5.57 |
| DimeNet++ | .044 | 32.6 | 24.6 | 19.5 | .0297 | .0230 | 7.56 | 6.53 | .331 | 6.28 | 6.32 | 1.21 | 0.98 | -5.67 |
| TFN† | .223 | 58 | 40 | 38 | .064 | .101 | - | - | - | - | - | - | - | - |
| SE(3)-Tr.† | .142 | 53 | 35 | 33 | .051 | .054 | - | - | - | - | - | - | - | - |
| EGNN† | .071 | 48 | 29 | 25 | .029 | .031 | 12 | 12 | .106 | 12 | 11 | 1.55 | 1.23 | -5.43 |
| PaiNN | .045 | 45.7 | 27.6 | 20.4 | .0120 | .024 | 7.35 | 5.98 | .066 | 5.83 | 5.85 | 1.28 | 1.01 | -5.85 |
| ET | .059 | 36.1 | 20.3 | 17.5 | .011 | .026 | 7.62 | 6.16 | **.033** | 6.38 | 6.15 | 1.84 | 0.84 | -5.90 |
| SphereNet | .046 | 31.1 | 22.8 | 18.9 | .0245 | .0215 | 7.78 | 6.33 | .268 | 6.36 | 6.26 | 1.12 | 0.91 | -5.73 |
| ComENet | .045 | 32.4 | 23.1 | 19.8 | .0245 | .0220 | 7.98 | 6.86 | .259 | 6.82 | 6.69 | 1.20 | 0.93 | -5.69 |
| SEGNN† | .060 | 42 | 24 | 21 | .023 | .031 | 15 | 16 | .660 | 13 | 15 | 1.62 | 1.08 | -5.27 |
| EQGAT | .053 | 32 | 20 | 16 | .011 | .024 | 23 | 24 | .382 | 25 | 25 | 2.00 | 0.86 | -5.28 |
| Equiformer | .046 | 30 | **15.4** | **14.7** | .0117 | .0230 | 7.63 | 6.63 | .251 | 6.74 | 6.59 | 1.26 | 0.70 | -5.82 |
| SAVENET-B | .039 | 24.8 | 18.4 | 16.3 | .0093 | .0227 | 6.64 | 5.43 | .058 | 5.48 | 5.43 | 1.18 | 0.69 | -6.04 |
| SAVENET-L | **.035** | 22.7 | 16.6 | 15.1 | **.0085** | **.0210** | **6.10** | **4.83** | .049 | **4.74** | **4.83** | **1.10** | **0.63** | **-6.14** |

of graphs. The three datasets include both invariant and equivariant targets for comprehensive evaluation of our proposed model. Further dataset details are provided in Appendix D. The rationale behind our design decisions and hyperparameters, as well as their specific values, are documented in Appendix F for reference.

**QM9**: QM9 dataset was used to evaluate the performance and efficiency of the models across twelve tasks for invariant target predictions. We present two distinct configurations of the SAVENET model, distinguished by their respective sizes: The SAVENET-B denotes the base variant, while the SAVENET-L is the larger model. We report various baseline models, including NMP [10], Schnet [28], Cormorant [1], LieConv [7], PhysNet [32], Dimenet [16], Dimenet++ [9], TFN [31], SE(3) Transformer [8], EGNN [27], PaiNN [29], ET [30], SphereNet [21], ComENet [33], SEGNN [3], EQGAT [18], and Equiformer [19].

**N-Body**: To assess the effectiveness of SAVENET with equivariant tasks, we conducted experiments on the extended N-Body dataset with harder targets [5]. The proposed SAVENET is compared with GNN, Tensor Field Networks [31], SE(3) Transformer [8], Radial Fields, PaiNN [29], EGNN [27], ClofNet [5], GCPNet [25]. Most of the baseline results are adopted from [5, 25]. The results for PaiNN are not reported in previous work [25] due to numerical stability. Notably, we incorporated vector activation functions into PaiNN's interaction layers to address this issue, allowing us to include PaiNN in the performance comparison.

**Molecule3D**: The Molecule3D dataset, possessing over $29\times$ the quantity of graphs as the QM9 dataset and an approximate $1.6\times$ and $1.9\times$ increase in the average number of nodes and edges respectively, is employed to evaluate the scalability of the model with respect to dataset size. For the Molecule3D dataset, we adopted the baselines as per [33]. As only the gap target has been reported in preceding works, we utilized the hyperparameters provided in [33] for evaluating additional targets. The details of the hyperparameters are described in Appendix F.1 The comparative baseline models for the Molecule3D include GIN-Virtual [33], Dimenet++ [9], Spherenet [21], and ComENet [33].

**Evaluation Metrics.** A set of metrics are used to measure the performance of the models. For the QM9 and Molecule3D datasets, each task is assessed using the Mean Absolute Error (MAE). For the N-Body dataset, the performance is measured by the Mean Squared Error (MSE) of the future position predictions. Given that each dataset encompasses multiple tasks, we employ two additional aggregate measures: standardized (std.) and logarithmic (log) metrics, to assess overall performance. As the error can be disproportionately influenced by a few outliers, we report the logarithmic error to prevent these outliers from dominating the evaluation.

**Efficiency Analysis.** To critically assess the efficiency of the models, we employ a set of metrics including training speed, model complexity, and memory consumption. All the measurements are conducted utilizing hyperparameters from the respective model authors, and the reference imple-

Table 3: Efficiency comparison of the proposed SAVENET with the baselines on QM9.

| Model Metric | std. MAE ↓ | log MAE ↓ | Batch Size ↑ | Memory GB ↓ | Latency ms ↓ | Samples/s ↑ | FLOPs G ↓ | Param. M ↓ | MACs G ↓ |
|---|---|---|---|---|---|---|---|---|---|
| DimeNet++ | 0.98 | -5.67 | 357 | 4.82 | 38.9 | 1646 | 30.15 | 1.89 | 15.08 |
| ComENet | 0.93 | -5.69 | 1174 | 0.96 | 26.4 | 2422 | 14.36 | 3.81 | 7.18 |
| SphereNet | 0.91 | -5.73 | 238 | 6.65 | 51.6 | 1240 | 31.80 | 1.89 | 15.90 |
| EQGAT | 0.86 | -5.28 | 980 | 1.74 | 53.3 | 1200 | 17.00 | **0.93** | 8.49 |
| Equiformer | 0.70 | -5.83 | 94 | 14.04 | 337.3 | 190 | - | 3.53 | - |
| SAVENET-B | 0.69 | -6.04 | **1660** | **0.85** | **23.1** | **2767** | **11.60** | 1.37 | **5.76** |
| SAVENET-L | **0.63** | **-6.14** | 580 | 2.48 | 50.6 | 1237 | 55.36 | 7.72 | 27.51 |

mentations are based on [18–20, 33]. To ensure consistent comparison, the experimental runs are executed on the same configurations utilizing an RTX 3090 GPU with 24 GB of graphics memory. The details of the efficiency analysis procedure are described in Appendix G and the theoretical time complexity is analyzed in Appendix E.

- *Training Speed.* To gauge the training speed of each model, we recorded the time consumed by a single forward/backward pass, expressed in milliseconds. Alongside this latency, we computed the number of samples that could be processed per second, employing similar measurement techniques.

- *Model Complexity.* The complexity of the models was evaluated by considering two key parameters: floating point operations (FLOPs) and multiply-accumulate operations (MACs). These metrics were computed utilizing the DeepSpeed library [23], with a consistent batch size.

- *Memory Consumption.* The memory consumption of each model was evaluated by assessing the amount of memory utilized across multiple batches. After processing each batch, the cache was cleared to ensure a realistic appraisal of memory usage. However, this practice of cache clearing significantly impedes the evaluation process, rendering a full dataset evaluation impractical.

## 5.1 Results

**Accuracy results on QM9.** Table 2 listed the experimental results of the QM9 dataset. SAVENET-B demonstrated superior performance on seven out of the twelve targets, surpassing all baseline models on these measures. While SAVENET-B did not yield the highest performance for certain parameters—namely, $\epsilon_{HOMO}$, $\epsilon_{LUMO}$, $C_\nu$, $R^2$, and ZPVE—it is noteworthy that it still delivered competitive results, indicating its robustness across diverse molecular properties. SAVENET-L further improved on SAVENET-B's performance, securing the best results on nine out of the twelve targets, with the exceptions of $\epsilon_{HOMO}$, $\epsilon_{LUMO}$, and $R^2$. Furthermore, in terms of the overall performance measures, SAVENET-L outperformed all other models, exhibiting the lowest standardized MAE and logarithmic MAE, reflecting its superior accuracy across all target properties. These results highlight the effectiveness of our proposed models, particularly SAVENET-L, in accurately predicting a wide range of molecular properties in the QM9 dataset.

Table 4: Performance comparisons on N-Body dataset.

| Method | ES(5) | ES(20) | G+ES(20) | L+ES(20) | std.MSE | logMSE |
|---|---|---|---|---|---|---|
| GNN | 0.0131 | 0.0720 | 0.0721 | 0.0908 | 0.0151 | -4.3876 |
| TFN | 0.0236 | 0.0794 | 0.0845 | 0.1243 | 0.0192 | -4.0978 |
| SE(3)-Tr. | 0.0329 | 0.1349 | 0.1000 | 0.1438 | 0.0252 | -3.8037 |
| Radial Field | 0.0207 | 0.0377 | 0.0399 | 0.0779 | 0.0110 | -4.6212 |
| PaiNN | 0.0158 | 0.0997[1] | 0.1029[1] | 0.1356[1] | 0.0835 | -2.5329 |
| ET | 0.1653 | 0.1788 | 0.2122 | 0.2989 | 0.0535 | -2.9587 |
| EGNN | 0.0079 | 0.0128 | 0.0118 | 0.0368 | 0.0044 | -5.6241 |
| ClofNet | 0.0065 | 0.0073 | **0.0072** | 0.0251 | 0.0029 | -6.0324 |
| GCPNET | 0.0070 | 0.0071 | 0.0073 | 0.0173 | 0.0024 | -6.1104 |
| SAVENET | **0.0062** | **0.0063** | 0.0082 | **0.0123** | **0.0020** | **-6.2268** |

**Efficiency results on QM9.** The results of efficiency experiments are presented in Table 3. We select the top five models based on their std. MAE performance for the efficiency comparison. Regarding efficiency, SAVENET-B stands out with its superior results, processing the highest number of samples per second and the lowest memory consumption. This demonstrates an impressive balance between performance and resource utilization. Additionally, SAVENET-B shows the lowest latency and number of FLOPs and MACs. The latency of SAVENET-B is $14.5\times$ lower compared to Equiformer, the previously most performant model with the lowest std. and log MAE. Even with a significant reduction in compute time, SAVENET-B surpasses the Equiformer in performance, illustrating efficiency advancements of SAVENET without performance compromise.

SAVENET-L manages to maintain a competitive efficiency despite its larger size. Its memory consumption is $2.6\times$ and $5.6\times$ times smaller compared to high-performing models such as SphereNet and Equiformer, respectively. Notably, SAVENET-L processes 1237 samples per second, which is comparable to the rates of SphereNet and EqGAT. Compared to Equiformer, the baseline model with the highest accuracy, SAVENET-L exhibits approximately $6\times$ lower latency. This suggests that SAVENET-L, even with its larger size, can maintain high performance without sacrificing speed.

**Results on N-Body dataset.** Table 4 presents the results of the N-Body systems. SAVENET outperformed all baseline methods on the ES(5), ES(20), and L+ES(20) tasks, whilst demonstrating competitive performance on the G+ES(20) task. Remarkably, SAVENET achieved the lowest average MSE and logarithmic MSE across all four tasks, illustrating its superior prediction precision for equivariant targets. These results emphasize the capability of our proposed model to accurately predict the future positions of particles in N-Body systems, even under challenging conditions where particles are under the influence of the gravity and force fields.

Table 5: Performance comparisons on Molecule3D dataset.

| Symmetry | Invariant | | | | | | Equivariant |
|---|---|---|---|---|---|---|---|
| Task | $\mu$ | $\varepsilon_{\text{HOMO}}$ | $\varepsilon_{\text{LUMO}}$ | $\Delta\varepsilon$ | std. MAE | logMAE | $\vec{\mu}$ |
| GIN-Virtual | 0.0882 | 0.0692 | 0.0632 | 0.1036 | 0.0592 | -2.8677 | 1.5233 |
| SchNet | 0.0532 | 0.0275 | 0.0265 | 0.0428 | 0.0263 | -3.6633 | 1.2082 |
| DimeNet++ | 0.0293 | 0.0240 | 0.0190 | 0.0306 | 0.0188 | -4.0139 | 1.2014 |
| SphereNet | 0.0288 | 0.0239 | 0.0183 | 0.0301 | 0.0184 | -4.0327 | 1.1836 |
| ComENet | 0.0345 | 0.0288 | 0.0252 | 0.0326 | 0.0220 | -3.8403 | 1.3521 |
| SAVENET-B | 0.0183 | 0.0190 | 0.0173 | 0.0290 | 0.0156 | -4.2257 | 0.0108 |
| SAVENET-L | **0.0136** | **0.0159** | **0.0143** | **0.0239** | **0.0128** | **-4.4408** | **0.0090** |

**Accuracy results on Molecule3D.** Our proposed models, SAVENET-B and SAVENET-L, outperformed several baseline models on the Molecule3D dataset. In terms of relative performance, SAVENET-B outperforms all tasks compared to other established models. SAVENET-B showed improved performance by approximately 37% on the $\mu$ task and 92% on the $\vec{\mu}$ task compared to the next best model, SphereNet. Our SAVENET-L model, further improved the performance of SAVENET-B. For the $\mu$ task, SAVENET-L showed an improvement of approximately 26% over SAVENET-B; for the $\vec{\mu}$ task, it showed an improvement of about 17%. Regarding aggregate measures, SAVENET-L also outperformed all other models, showing a relative improvement of around 18% in the std. MAE metric and 5% in the log MAE metric compared to SAVENET-B. These results demonstrate the superior performance and scalability of our proposed models, especially SAVENET-L, on the Molecule3D dataset.

**Efficiency results on Molecule3D.** The results of the efficiency evaluations on the Molecule3D dataset are illustrated in Table 6. Impressively, SAVENET-B continues to dominate with respect to efficiency, manifesting the highest sample processing rate and minimal memory consumption. This shows an exemplary trade-off between accuracy and resource management.

---

[1]The results are reported after adding vector activation functions. These results were initially unavailable due to numerical instability.

Table 6: Efficiency comparison of the proposed SAVENET with the baselines on Molecule3D.

| Model Metric | std. MAE ↓ | log MAE ↓ | Batch Size↑ | Memory GB ↓ | Latency ms ↓ | Samples/s ↑ | FLOPs G ↓ | Param. M ↓ | MACs G ↓ |
|---|---|---|---|---|---|---|---|---|---|
| DimeNetPP | 0.0188 | -4.0139 | 62 | 12.14 | 84.2 | 704 | 70.53 | 1.89 | 35.27 |
| SphereNet | 0.0184 | -4.0327 | 43 | 17.32 | 134.6 | 448 | 92.56 | 1.90 | 46.28 |
| ComENet | 0.0220 | -3.8403 | 277 | 3.37 | 40.7 | 1536 | 47.25 | 7.36 | 23.63 |
| SAVENET-B | 0.0156 | -4.2257 | **329** | **2.84** | **27.8** | **2304** | **15.33** | **1.05** | **7.63** |
| SAVENET-L | **0.0128** | **-4.4408** | 159 | 5.37 | 51.9 | 1216 | 38.93 | 3.01 | 19.37 |

## 5.2 Ablation Study

Table 7 presents the results of our ablation study on the proposed SAVENET model, where we investigate the impacts of Direction Noise (DN), Vector Activations (VA), and varying the number of layers (L) on the QM9 dataset. In Table 7, we report standard Mean Absolute Error (std. MAE), the percentage change in std. MAE, and the logarithmic Mean Absolute

Table 7: Ablation study of SAVENET

| Model | L | std. | Δstd. | log |
|---|---|---|---|---|
| SAVENET - DN - VA | 6 | 0.79 | - | -5.85 |
| SAVENET - VA | 6 | 0.77 | -2.5% | -5.95 |
| SAVENET - VA | 8 | 0.71 | -10.1% | -5.98 |
| SAVENET | 8 | 0.69 | -12.7% | -6.04 |
| SAVENET | 12 | 0.63 | -20.3% | -6.14 |

Error (logMAE), respectively. The importance of DN is illustrated in the first two rows, where the std. MAE is reduced by 2.5%. Moreover, scaling the model with DA from six to eight layers yields a 7.8% reduction in error. Introducing VA into an eight-layer model further improves the performance by 2.8%. Lastly, expanding the model to 12 layers offers a substantial performance improvement of 8.7%.

## 6 Conclusion, Limitations and Future Works

This paper has focused on learning geometric representations of molecular structures. We introduced SaVeNet, a scalable message-passing neural network designed for learning geometric representations from 3D graphs. Furthermore, we implemented a unique approach to integrate geometric features, capturing complex geometric relationships within 3D structures without resorting to computationally complex operations. The performance of this framework was demonstrated on both synthetic and real-world datasets. For the latter, SaVeNet was evaluated across various core tasks in molecular representation learning. When compared with state-of-the-art methods, the effectiveness and efficiency of our proposed architecture were clearly validated. Our comprehensive examination of network efficiency and our comprehension of the trade-offs between expressiveness and efficiency will provide a foundation for future advancements in molecular representation learning. However, like many studies in this area, our work does have limitations. The primary constraint, shared with the baseline model discussed in this paper, is the dependency on known geometric structures of molecules at their equilibrium state. In practice, obtaining accurate 3D confirmations can be costly. An intriguing direction for future research is to explore an end-to-end framework that employs sequential representations of molecules, such as SMILES strings [34], to generate their 3D geometry, where the predicted 3D geometry could then be used to model molecular properties. Another promising avenue is to investigate the performance of larger variations of SaVeNet. Furthermore, assessing the impacts of efficiency improvements on larger models, potentially achieved through shared parameters in various configurations, is a worthwhile endeavor. We anticipate that our findings will pave the way for future investigations in this domain, potentially leading to even more efficient and effective modeling techniques.

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

# Appendix

## A  Proof of Lemma 1

**Lemma 1:**  Consider a known geometric graph with at least one node, and assume that each node in this graph has at least one connection. Let a new node $j$ be added to this graph, stipulating that $j$ is connected to at least one existing node. Then, the position $p_j$ of node $j$ can be determined in constant time using the $(p_0, r_{ij}, d_{\vec{ij}})$ properties.

*Proof of Lemma 1:* In our context, we are dealing with a geometric graph whose node properties are known a priori. When a new node $j$ is appended to the graph, such that it is connected to at least one node, we can select any node $i \in \mathcal{N}_j$ from the neighbors of $j$ to establish the position $c_j$ of node $j$.

Proceeding by contradiction, assume that multiple feasible positions $p_j$ could satisfy the conditions $||c_i - c_j|| = r_{ij}$ and $\vec{d}_{ij}$ for a given position $p_i$. In multi-dimensional spaces, including 3D, there exist multiple positions satisfying the distance constraint $||c_i - c_j|| = r_{ij}$. Specifically, any point lying on a sphere in 3D space, with $c_i$ as the center and $r_{ij}$ as the radius, could satisfy this condition. Yet, there is only one such position that can satisfy the direction vector $\vec{d}_{ij}$ from $c_i$ to $c_j$. This fact contradicts our assumption, and hence concludes the proof.

## B  Proof of Theorem 1

**Theorem 1:**  Assuming the input graph is strongly connected, the input space of SAVENET , defined as $(r, d)$, is capable of reconstructing the rotation-equivariant geometric structure of the input graph. The reconstructed graph maintains the relative spatial configuration of nodes and edges in the input graph, the translation transformation is required for exact alignment with the original graph.

*Proof of Theorem 1:* By the definition of a strongly connected graph, for any given node $i$, there exists a path $\Pi_i^j$ to every other node $j$. Hence, we can arbitrarily choose any node from the graph as a reference for reconstructing the rotation-equivariant geometric structure of the input graph. Let's select node $i$ for this purpose. Since $c_i$ is arbitrary, we can conveniently set it at the origin without loss of generality.

Utilizing Lemma 1, we can identify the relative position of every other node $j \in V$ by traversing each edge on the path $\Pi_i^j$ from $i$ to $j$. In essence, the relative position of a node $j$ with respect to node $i$ can be computed by the accumulation of relative positions along the path $\Pi_i^j$.

Assuming, by way of contradiction, that multiple relative positions of node $j$ with respect to node $i$ exist. However, this is counter to Lemma 1, which affirms that for given $c_i, r_{ij}, \vec{d}_{ij}$, there exists a unique location. This principle can be extrapolated to any number of nodes in the graph. Further, as we can ascertain the position of a node after it's added to the reference graph, we can similarly determine the position of any subsequent node $k$ relative to $V \cup \{j\}$.

Thus, we conclude that the input graph's rotation-equivariant geometric structure can be reconstructed using $(d, r)$ properties, thereby completing our proof.

## C  Proof of Theorem 2

**Theorem 2:**  The equivariant representation of SAVENET is equivariant to any given rotation matrix in $R \in SO(3)$. The invariant representation of SAVENET is invariant to any given transformation matrix in SE(3).

*Proof of Theorem 2:* Our proof unfolds in two stages: first, we discuss the symmetrical properties of SAVENET in the input space. Second, we extend these properties to the entire network.

*Inputs.*  The input to SAVENET consists of direction vectors, denoted $\vec{\beta}$. It's clear that these vectors, representing direction, maintain their properties under any global rotation. Formally, if $f_b(\mathcal{G})$ represents the function that extracts $\vec{\beta}$ from an input graph $\mathcal{G}$, we have that $R f_b(\mathcal{G}) = f_b(R\mathcal{G})$ for all

$R \in SO(3)$. Moreover, any noise introduced during initialization decays over time until reaches zero before the end of the training phase. Therefore, having no effect on the inference process.

The invariant input vectors of SAVENET, which represent atomic numbers and interatomic distances, are clearly invariant under any transformation in $SE(3)$ because these properties do not depend on the geometrical information of the system. Hence, atomic numbers remain constant under global rotations and translations. Similarly, the distances between atoms, being local properties, remain unchanged under global transformations.

*Network Propagation:* Having established the symmetries at the input level, we extend our analysis to the entire network of SAVENET. Two critical components are *Inter-atomic Interactions* and *Atom-wise Blocks*, which we shall now scrutinize.

*Inter-atomic interactions.* The interaction function $e$ can be dissected to analyze its operations on the equivariant and invariant branches. Here, $\phi_b$, $\phi_d$, and $\phi_v$ denote the vector update layer, the linear transformation, and the multiplication of $v$ representation with the linear transformation of $s_j$, respectively. In the equivariant branch, operations can be expressed as:

$$e_v(s_j, V_j, r_{ij}, \vec{\beta}_{ij}) = \phi_b(\vec{\beta}_{ij}) \odot \phi_d(s_j) \odot \eta_d(r_{ij}) + V_j \odot \phi_v(s_j) \odot \eta_r(r_{ij}) \qquad (7)$$

Each component in the above expression preserves symmetry properties. More specifically, the first term $\phi_b(\vec{\beta}_{ij})$ preserves equivariance because the same transformation is applied across spatial dimensions. The second term, $\phi_d(s_j)$, is a linear transformation applied to an invariant, and therefore, remains invariant under any transformation in $T$. The third term, $\eta_d(r_{ij})$, which multiplies a basis function and a cutoff function for the distance $r_{ij}$, does not change under transformations in $T$. Thus, it is also invariant under $T$. The fourth term, $\phi_v(s_j)$, involves a multiplication operation of $v$ representation with a linear transformation of $s_j$, hence, it produces an equivariant vector since the linear transformation does not affect the spatial dimensions. Finally, the term $\eta_r(r_{ij})$ represents an invariant scalar, remaining unchanged under transformation.

The first part $\phi_b(\vec{\beta}_{ij}) \odot \phi_d(s_j) \odot \eta_d(r_{ij})$ involves the Hadamard product of an equivariant vector ($\phi_b(\vec{\beta}_{ij}) \in \mathbb{R}^{C \times 3}$) with invariant scalar values ($\phi_d(s_j), \eta_d(r_{ij}) \in \mathbb{R}^C$). Hadamard product operates element-wise and does not alter the spatial dimensions, hence the resulting vector preserves the equivariant properties. The second part $V_j \odot \phi_v(s_j) \odot \eta_r(r_{ij})$ involves the Hadamard product of an equivariant vector ($V_j \in \mathbb{R}^{C \times 3}$) with an invariant scalar representations. Again, the resulting vector is equivariant since the same values are multiplied across spatial dimensions. Finally, the addition operation in $e_v$ combines two equivariant vectors. As the addition of equivariant vectors preserves equivariance, the result will be equivariant under any rotation matrix $R \in SO(3)$. This concludes our examination of the equivariance of the interaction function $e_v$ under any rotation $R \in SO(3)$. Similarly, the operations in the invariant update branch can be captured as:

$$e_s(s_j, v_j, r_{ij}, \vec{\beta}_{ij}) = \phi_s(s_j) \odot \eta_s(r_{ij}) \qquad (8)$$

where $\phi_s$ denotes a linear transformation, and $\eta_s$ represents a transformation applied using basis and cutoff functions. Both these functions are invariant under any transformation in $T$.

The updated $s$ and $V$ representations, determined for each adjacent node, are subsequently aggregated through summation. Given that summation inherently upholds symmetry characteristics, the procedures employed in the *Inter-atomic interactions* layer maintain the equivariant nature of $V$ for all rotations $R$ in $SO(3)$, as well as the invariant nature of $s$ under any transformation $T$ in $SE(3)$.

*Atom-wise Blocks.* Assuming that the inputs to this block are invariant for scalar representations and equivariant for vector representations, we can extend our proof to the operations within this block. The operations in the invariant update branch can be encapsulated as follows:

$$\phi_m(||\phi_{vu}(V)|| \cup s) \qquad (9)$$

In this equation, $\phi_{vu}$ denotes the vector update function which is, as previously established, equivariant under any rotation matrix $R$. The norm operation $|| \cdot ||$ applied to $V$ is invariant, as the magnitude of the vector does not change with rotation. Consequently, the composition $||\phi_{vu}(V)||$ is invariant to global rotations. Furthermore, applying the linear transformation $\phi_m$ to the concatenation of two invariants, in this case $||\phi_{vu}(V)||$ and $s$, remains invariant under any transformation in $SE(3)$. The operations in the equivariant branch of the *Atom-wise blocks* can be described as:

$$\text{VA}(\phi_{vu}(V) \odot \phi_v(||\phi_{vu}(V)|| \cup s_i)) \qquad (10)$$

This equation is similar to the one describing the invariant update branch, with a difference lying in the linear transformation $\phi_v$ that uses different learnable parameters. Hence, the second term is invariant. The first term, $\phi_{vu}(V)$, is equivariant as it is a vector update function applied to $V$. The Hadamard product of this equivariant representation with the invariant term preserves its equivariance under any rotation $R \in SO(3)$. In conclusion, the *Atom-wise blocks* preserve the symmetrical properties for both $(s, V)$ representations. The proof for Theorem 2 is thus complete.

## D  Datasets

The three datasets used in this study evaluate our proposed SAVENET on three tasks in geometric representation learning over both synthetic and real-world datasets. In addition to their diversity in their real-world use cases, these tasks demonstrate a rich diversity not just in their practical implications, but also in their requisite output specifications. In other words, they demand the prediction of both invariant and equivariant targets, each necessitating different computational considerations. This multi-faceted evaluation strategy, which includes varying input data types and output requirements, provides a comprehensive view of the model's capabilities, and furthers the understanding of its strengths and potential areas for refinement. By analysing the model's performance across these distinct scenarios, we aim to exhibit SAVENET 's adaptability and potential as a valuable tool in advancing the field of geometric representation learning.

**QM9.** Widely adopted small molecule property prediction dataset. It consists of 131,831 stable molecules with nine heavy atoms. We utilize the established split from [16], a standard in subsequent works [9, 21, 33]. QM9 contains twelve tasks that involve various aspects of molecular properties. The multi-task nature of the dataset demonstrates the effectiveness of the evaluated models.

**Newtonian many-body system.** Newtonian many-body system is a synthetic dataset that is generated based on particle simulations. The task is to predict the future positions of the particles based on the initial location and velocity. The performance was measured with the mean square error (MSE) metric between predicted and ground truth. Recently the existing work by [27] extended by increasing the particle size and introducing various force fields. By introducing the force fields, the existing task became more difficult. The initial work evaluated the models' performance based on the future position on 5-body (ES(5)). More challenging tasks to predict future locations of the 20-body (ES(20)) task is introduced in [5]. Furthermore, complex conditions integrated with gravity field (G+ES(20)) and Lorentz-like force field (L+ES(20)). The performances of the models were evaluated under challenging conditions. The hyper-parameters of our work SAVENET were chosen based on the best-performing hyper-parameters on ES(5) based on the best validation score.

**Molecule3D.** This dataset comprises 3,899,647 molecules, making it approximately 29.5 times larger than the QM9 dataset, providing a significant platform to assess the scalability of our model. Predefined splits are utilized, based on random sampling as detailed in [37]. Though Molecule3D has fewer targets than QM9, with only four targets including dipole moment vector ($\vec{\mu}$), energies of the highest occupied and lowest unoccupied molecular orbitals ($\epsilon_{\text{HOMO}}$, $\epsilon_{\text{LUMO}}$), and orbital energy gap ($\Delta\epsilon$), it offers an essential testbed for model evaluation. The dipole magnitude ($\mu$) can be easily derived from $\vec{\mu}$, highlighting the significance of equivariance in models due to the influence of global rotations on $\vec{\mu}$. For a fair comparison, we have reported standardized MAE for invariant properties.

## E  Theoretical complexity analysis

In the following section, we present a detailed theoretical complexity analysis of our proposed model and baselines, which is summarized in a comprehensive table for understanding the model's computational efficiency. In the table, we provide a breakdown of the computational com-

Table 8: Complexity analysis.

| Complexity | Dimenet++ | SphereNet | ComENet | SAVENET |
|---|---|---|---|---|
| Initialization | $n^3$ | $n^4$ | $n^2$ | $n^2$ |
| Encoder | $l_e n^3 d^2$ | $l_e n^3 d^2$ | $l_e n^2 d^2$ | $l_e n^2 d^2$ |
| Decoder | $l_d n d^2$ | $l_d n d^2$ | $l_d n d^2$ | $l_d n d^2$ |
| Overall | $n^3 d^2$ | $n^3 d^2$ | $n^2 d^2$ | $n^2 d^2$ |

plexities associated with the main components of the model's architecture. This includes complexities during the graph embedding phase, the encoder, and the decoder phase. By presenting this theoretical complexity analysis, we aim to provide an in-depth understanding of the operational demands of our model and highlight its computational advantages.

# F   Implementation details

The reference implementation can be found in https://github.com/EfficientGraphs/SaVeNet.

## F.1   Hyperparameters

The hyperparameter search space of SAVENET-B on QM9 and Molecule3D is shown in Table 9. The final hyperparameters are chosen based on the hyperparameters with the best performance on the validation set of each dataset. The hyperparameters of SAVENET-L is determined based on the best hyperparameters found for SAVENET-B on the validation set. Then the hidden dimensions and the number of layers are scaled on SAVENET-L. On the QM9 dataset, SAVENET-L has two times more hidden dimensions and decoder hidden dimensions compared to SAVENET-B. SAVENET-L utilized 256 hidden dimensions compared to 128 dimensions, and SAVENET-L has 1.5 more layers increasing from eight layers to twelve layers. On the other hand for the Molecule3D dataset, hidden dimensions are increased 1.5 times from 128 to 192 and the number of layers increased to eight layers from six layers. The hyperparameter search space of SAVENET on N-Body is shown in Table 10.

Table 9: Hyperparameter search space of SAVENET-B on QM9 and Molecule3D.

| Hyperparameters | QM9 | Molecule3D |
|---|---|---|
| Epochs | 1000 | 300 |
| Batch size | 32, 64, 128 | 64, 128, 256 |
| Number of layers | 6, 8 | 4, 6 |
| Hidden dimensions | 64, 128 | 128, 192 |
| Decoder dimensions | 128, 256 | 256, 384 |
| Number of RBF | 12, 16, 24 | 12, 16, 24 |
| Base Learning Rate | 1e-3, 5e-4, 2e-4 | 1e-3, 5e-4, 2e-4 |
| Learning rate decay | 0.6, 0.8, 0.9 | 0.6, 0.8, 0.9 |
| Learning rate patience | 10, 15, 20 | 10, 15, 20 |
| Vector dropout | 0.0, 0.1, 0.25 | 0.0, 0.1, 0.25 |
| Direction noise decay | 150, 250 | 150, 250 |

Table 10: Hyperparameter search space of SAVENET-B on N-Body.

| Hyperparameters | N-Body |
|---|---|
| Epochs | 500 |
| Number of layers | 4 |
| Hidden dimensions | 128 |
| Decoder dimensions | 128, 256 |
| Number of RBF | 12, 16, 24 |
| Base Learning Rate | 1e-3, 5e-4, 2e-4 |
| Learning rate decay | 0.6, 0.8, 0.9 |
| Learning rate patience | 10, 15, 20 |
| Vector dropout | 0.0, 0.1, 0.25 |
| Direction noise decay | 150, 250 |

# G   Efficiency evaluation

In this section, we conduct a rigorous evaluation of our proposed model, employing a set of carefully devised efficiency metrics. These metrics are instrumental in providing a comprehensive understanding of the model's operational efficacy and practical viability.

The key metrics used include:

**Training Speed**: This metric is crucial in assessing the efficiency of the model from a computational perspective. It gauges the rate at which the model processes data and adjusts parameters during training, which is particularly pertinent in large-scale applications. For the purpose of deriving accurate

results, each model underwent a warm-up phase of 5,000 steps before we timed the forward/backward passes across the entire training dataset. The training dataset was deliberately left unshuffled to maintain consistent batches, thereby facilitating efficient and comparable measurements. The final result was then determined by calculating the average time spent on these passes.

**Model Complexity**: A model's complexity often directly correlates with its performance. However, a balance between complexity and computational efficiency is desirable. Hence, we consider this metric to quantify the intricacy of our model's architecture and algorithms, which also informs us about its interpretability and potential for scaling. Mirroring the approach adopted during the training speed experiment, we maintained uniform batches throughout the entire training dataset, and the resultant metrics were averaged upon the conclusion of an epoch. The metrics were computed with a consistent batch size of 64 to accommodate memory-demanding models.

**Memory Consumption**: The amount of memory consumed during the model's operation is a significant factor, especially for deployment in resource-constrained environments. This metric provides insights into the model's memory requirements, both during the training phase. Since evaluating the complete dataset evaluation was impractical, we employed an effective measurement strategy. To accommodate the impact of various sample sizes on memory usage, we constructed a worst-case scenario, selecting batches containing the largest samples in a given dataset. An efficient strategy for identifying these substantial samples involved considering the number of nodes in each sample. Consequently, we extracted the $n$ largest samples from the dataset, where $n$ is equivalent to the batch size. The memory consumption was then repeatedly measured for this batch, and the results were averaged to yield the final value.

