# OpenReview forum: "SaVeNet: A Scalable Vector Network for Enhanced Molecular Representation Learning"
_NeurIPS.cc/2023/Conference — NeurIPS 2023 poster_

### Official Review · Reviewer_FRcS · 2023-06-20

**Soundness:** 4 excellent
**Presentation:** 2 fair
**Contribution:** 3 good
**Rating:** 5
**Confidence:** 4

**Summary:**

This paper introduces a novel molecule representation network that enhances the learning capacity and scalability through the integration of innovative initialization techniques and activation functions for vector features. The conducted experiments validate the network's proficiency in three distinct molecule representation tasks, demonstrating its capability to effectively learn from data while maintaining scalability.

**Strengths:**

- A new equivariant network capable of learning scalar and vector features.
- Superior performance and scalability compared to previous methods.


**Weaknesses:**

- The presentation of the method can be improved. The symbols and equations employed are unclear, such as the presence of a comma in Eq. 4 and the excessive use of unnecessary new symbols in Eq. 5. Furthermore, the substitution of the symbol for multiplication with the letter 'x' in line 187 could be clarified. Additionally, it is suggested that the authors consider using a figure to illustrate the message passing of scalar and vector features, similar to previous methods like DimeNet.

- The core difference between the proposed network and previous networks for vector features, such as PaiNN, remains unclear. While it appears that the primary distinction lies in the different processing of features, it would be beneficial to highlight and discuss this core difference in greater detail.

**Questions:**

Regarding the direction noise, it would be helpful to clarify whether theta and phi are deterministic functions of the node type. If they are, applying a rotation to the graph would not change the node type, thereby maintaining the invariance instead of the equivariance of VEN(F_i). So why can VEN(F_i) be an equivariant vector initialization?  Further explanations and intuitions could be added to address this point and enhance understanding.

---

> ### Author Rebuttal · Authors · 2023-08-10
>
> Dear Reviewer FRcS,
>
> Thank you for your thorough assessment of our work. We value your feedback and would like to address your concerns and suggestions as follows:
>
> > Presentation and Equation Clarifications
>
> 1. **Eq. 4 Ambiguity**: We've refined Eq. 4 to clarify the scalar-vector tuple representation.
> $$\begin{split}
> e_s(s_j, V_j, {r}\_{ij}, \vec{\beta}\_{ij}) =& \phi_s(s_j) \odot \eta\_s(r\_{ij}) \\\\
> e_v(s_j, V_j, {r}\_{ij}, \vec{\beta}\_{ij}) =& \text{VA}\big(\phi\_b(\vec{\beta}\_{ij}) \odot \phi_d(s\_j) \odot \eta_d(r\_{ij})  + V_{j} \odot \phi\_v(s\_j) \odot \eta_r(r\_{ij})\big)\end{split}$$
> 2. **Concerns about Eq. 5**: We concur that minimizing the use of extraneous symbols is crucial for clarity. Hence, we've revisited Eq. 5 and streamlined the notation. We've also ensured that the scalar and vector interaction pathways are defined with precision, offering coherence with the revised Eq. 5.
> $$\begin{split}
> \text{IA}(s, V, r, \vec{\beta})  &=  (s\_i, V\_i) + \sum\nolimits\_{j\in\mathcal{N}\_i}{e(s\_j, V\_j, {r}\_{ij}, \vec{\beta}\_{ij})} = (s'\_i, V'\_i) \\\\
> s'\_i &= s\_i + \sum\nolimits\_{j\in\mathcal{N}\_i}{e\_{s}(s\_j, V\_j, {r}\_{ij}, \vec{\beta}\_{ij})} \\\\
> V'\_i &= V\_i + \sum\nolimits\_{j\in\mathcal{N}\_i}{e\_{v}(s\_j, V\_j, {r}\_{ij}, \vec{\beta}\_{ij})}\end{split}$$
> 3. **Symbol for Multiplication in Line 187**: Thank you, this has been corrected as '$\times$'.
> 4. **Illustrative Figure for Message Passing**: We have incorporated an illustrative figure in uploaded PDF Figure 2.
>
> > The difference between SaVeNet and previous networks
>
> We'd like to elucidate on the core differences as follows:
> - **Embedding:** Our approach to feature embedding is fundamentally novel. Rather than relying on fewer 3D features, our goal is to craft embeddings that relay lossless information for the 3D molecular graph and resonate with SaVeNet's design. To illustrate, while several existing methods, including Spherenet, ComENet, PaiNN, ET, and EQGAT, tap into diverse features like distance, torsion angle, and directional vectors, our unique feature sets, detailed in Section 3.1, set us on a distinct trajectory right from inception. Furthermore, we've provided both theoretical (Theorem 1) and empirical validation, underpinning the efficacy of our representations in determining the 3D structure.
> - **Initialization:** Contrary to conventional practice in existing works, where vector initializations are often nullified, making the network intrinsically derive these representations, our method sidesteps this potential computational pitfall. Our initialization scheme is tailored to be both efficient and resource-conscious.
> - **Architectural Innovations in Message Passing**: SaVeNet introduces a unique message-passing paradigm, elaborated in Sections 3.1 and 3.2. We've appended Figure 2 as suggested. Our design incorporates our proposed vector activation techniques, which not only delineates our model from predecessors but also fortifies its reliability and scalability. To provide perspective, while models like PaiNN were conceived primarily for molecular property prediction, their inherent design may not extend seamlessly to more diverse tasks. A case in point is PaiNN's convergence issues on certain demanding targets [1]. Our proposed vector activation function not only alleviated PaiNN's limitations but also enriched its capabilities, as showcased in Table 4's performance comparison.
> - **Comparative Performance Analysis**:Our evaluations underscore SaVeNet's superiority, which was evident when models like PaiNN and ET were benchmarked against N-Body tasks. The results clearly attest to the innovations and the efficacy SaVeNet brings to computational molecular science.
>
> In summary, from the initial motivations, through feature design and architecture, to real-world application performances, SaVeNet has been meticulously crafted to push the boundaries of what's possible in the domain, while addressing the limitations of its predecessors.
>
> > Regarding the direction noise
>
> Let us provide clarity on the concerns raised:
> 1. **Initialization Concerns with \($\theta\$) and \($\phi$\)**:
>     - Both are intrinsically linked to node types. This design choice is deliberate, taking into account the potential noise from simulations, e.g., DFT. By associating noise with atomic numbers, our model achieves robustness against variances in node coordinates.
>
> 2. **Alternative Vector Initialization Techniques**:
>     - **Direction to the Center of Mass (MCM)**:
>       $$\text{MCM}(i, G) = \sum_{j \in \mathcal{G}}{\frac{c_{j}}{N}} - c_{i}$$
>       Demonstrably equivariant under rotations as $$\text{MCM}(i, RG) = \sum_{j \in \mathcal{G}}{\frac{Rc_{j}}{N}} - Rc_{i}$$.
>
>     - **Direction to Neighborhood Center of Mass (NCM)**:
>       $$\text{NCM}(i, G) = \sum_{j \in \mathcal{N_{i}}}{\frac{c_{j}}{|\mathcal{N_{i}}|}} - c_{i}$$
>       Provides a localized node representation.
>
>     - **Direction to the Nearest Node (NN)**:
>       $$\text{NN}(i, G) = {c_{j} - c_{i} \quad | \quad c_{j} = \text{argmin}_{k\in \mathcal{N}} \left( ||c_k - c_i|| \right) }$$
>       Recognizes the invariance of the closest node under rotations.
>
> 3. **Empirical Results on Molecule3D**:
> | Method   | Homo   | Gap    |
> |----------|--------|--------|
> | MCM      | 0.0213 | 0.0321 |
> | MCM + DN | 0.0208 | 0.0313 |
> | NCN      | 0.0213 | 0.0327 |
> | NCN + DN | 0.0199 | 0.0301 |
> | NN       | 0.0211 | 0.0321 |
> | NN + DN  | 0.0205 | 0.0318 |
> | DN       | 0.0190 | 0.0290 |
>
> This table elucidates the relevance of introducing directional noise. Across initialization techniques, the integration of DN consistently elevates model performance.
>
> We're grateful for your insightful comments and hope that these amendments will address your concerns effectively.
>
> [1] A. Morehead and J. Cheng, ‘Geometry-Complete Perceptron Networks for 3D Molecular Graphs’, _AAAI Workshop on Deep Learning on Graphs: Methods and Applications_, 2023.

---

> ### Author Response · Authors · 2023-08-21
> **Response to reviewer FRcS**
>
> Thank you for taking the time to provide insightful comments. We are glad that you recognize the soundness of our work as `excellent` and our contribution as `good`. We revised our paper based on your suggestions to improve the `presentation` of our work and addressed your questions in our previous response.  Please let us know if there are any additional questions or comments, we would be more than happy to address them. Thank you!

---

### Official Review · Reviewer_jP4K · 2023-06-28

**Soundness:** 3 good
**Presentation:** 2 fair
**Contribution:** 3 good
**Rating:** 6
**Confidence:** 3

**Summary:**

This paper proposes an effective and efficient equivariant graph neural network for geometric learning on molecules. The model encodes 3D graphs with node types and coordinates and outputs scalar and vector representations. The message passing process is purely scalar-based, which enjoys more efficiency than baselines using irreducible representations and high-order tensor objects . A vector initialization based on spherical coordinate system is utilized for better numerical stability. Experiments on QM9, N-body and molecule3D demonstrate the superiority of the proposed model over the baselines.

**Strengths:**

1. The proposed scalar-based message passing scheme enjoys theoretically better efficiency and empirically better performance compared to existing baselines.
2. The experiments are thorough, including three popular benchmarks and a variety of baselines.
3. The empirical analysis on efficiency is comprehensive, covering  time for forward/backward pass, FLOPs, MACs and memory consumption.

**Weaknesses:**

1. Several points claimed in the contribution are not well supported:
   - The authors state that the vector initialization can ensure numerical stability and speed up model convergence. However, no theoretical analysis on why it can ensure numerical stability is provided, and neither experimental evidence nor theoretical proof of speeding up convergence is provided.
   - The scaled version of the baselines should also be compared with SAVENET-L to show that the proposed model benefits more from the scaling (i.e. Has better scalability). Moreover, it is not shown whether the scalability of the proposed model can hold when keeping increasing the model size.
2. The presentation of the model structure can be optimized (e.g. add an overview figure).

**Questions:**

1. What is the $h_j$ in Eq.5?
2. What do $n, l_e, l_d$ represent in the theoretical complexity analysis?
3. Are the coordinates fixed during message passing?
4. Why can the directional initialization improve numerical stability? What is the reason for using spherical coordinate system as initialization? What is the difference if we use random vectors as initialization and anneal their norm to zero during training?
5. I think the computational scalability of GNNs is mainly restricted by the fully connected message passing which has quadratic complexity with respect to the number of nodes. Therefore in large graphs (e.g. proteins), people often use k-nearest neighbors to reduce the complexity to be linear to the number of nodes. However, the expressiveness of scalar-based equivariant GNNs can only be ensured in nearly fully connected message passing [1]. So can you show that the proposed models can maintain its superiority over the baselines when adapted to larger graphs like proteins, where edges might have to be much more sparser than in fully connected graphs. Or, under the current experiment settings, will the superiority be maintained if the edges get sparser (e.g. use k-nearest neighbors for message passing and decrease the K or decreasing the cutoff in the radial graph)?
6. Will the scalability holds when keeping increasing the hidden size and the number of layers?

[1] Scalars are universal: Equivariant machine learning, structured like classical physics

**Limitations:**

When scaling to larger graphs (e.g. proteins), the edges may have to be sparser (e.g. K-nearest neighbors) and the scalar-based message passing potentially will have worse expressiveness.

---

> ### Author Rebuttal · Authors · 2023-08-10
>
> Dear Reviewer jP4K,
>
> Firstly, we extend our gratitude for the careful review and insightful feedback on our manuscript. We address each comment in detail to offer a clearer understanding of our work.
>
> > Numerical stability and model convergence
>
> 1. **Relevance of Vector Initialization**: Recent works such as SchNet, DimeNet, SphereNet, and ComENet have established that enhancing the richness of features, can significantly boost performance in molecular tasks. Initializing with vectorial features, as opposed to empty or sparse values, provides the network with a richer context from the outset. This better informed start allows the model to begin its learning process from a more advantageous position.
> 2. **Stability and Convergence**: PaiNN's challenges with the N-Body dataset highlight instability issues seen in some models. Rather than simply failing to converge, these models can have error spikes during training due to numerical instability. This often arises in networks using vector representations, especially when merging vector and scalar values, like in architectures such as GVP, PaiNN, ET, and EQGAT. If these networks perform vector-scalar combinations too early, before directional data is introduced, they can become unstable, as observed with PaiNN. This is particularly problematic when the sum of neighboring vectors is close to zero, leading to gradient explosions. Hence, initiating robust representations is pivotal, especially when early vector-scalar operations are involved.
>
>
> > Scaling the baselines
>
> To address this question, we conducted experiments on the QM9 dataset to scale high-performing baselines to investigate their scalability compared to SaVeNet-L.
>
> **1. Baseline Selection**:
> We intentionally chose to scale both vectorial and spherical representation models to provide a broad-based comparison.
> - **ET**, which excels in performance with aggregated std. MAE and log MAE metrics, representing vectorial models.
> - **ComENet** was chosen for its efficiency advancements over SphereNet, representing spherical models.
>
> **2. Metrics and Findings from the Comparison**:
>
> From our experiments below, a few key insights emerge:
>
> |Model|std.|log|Batch|Memory|Latency|Samples/s|FLOPs|Param.|MACs|
> |---|---|---|---|---|---|---|---|---|---|
> |ET|0.84|-5.90|418|3.66|49.5|1280|42.98|6.86|21.40|
> |ET-L|0.84|-5.96|287|5.33|74.9|832|63.36|10.00|31.55|
> |ComENet|0.93|-5.69|1174|0.96|26.4|2368|14.36|3.81|7.18|
> |ComENet-L|0.97|-5.74|748|4.45|35.7|1728|39.50|11.30|19.75|
>
> Observing the comparison, it's evident that SaVeNet not only demonstrates superior performance with the base model but also capitalizes on its design to scale effectively, as reflected in SaVeNet-L's results.
>
> > Illustration of the model structure
>
> As suggested, we uploaded PDF Figure 2 for an illustration.
>
> > What is $h_j$ in Eq.5?
>
> The $h_j$ denotes the latent representation of a node $j$ computed with a linear transformation without bias to scalar representations $s$ . The linear transformation is applied to distinguish between a source node $j$ a destination node $i$. However, recognizing that the transformation functions $\phi_s$ and $\phi_d$ can encapsulate this linear transformation, we decided to simplify our equation by omitting $h_j$. The revised version of the equations are as follows:
> $$\begin{eqnarray}
> e_s(s_j, V_j, {r}\_{ij}, \vec{\beta}\_{ij}) &=& \phi_s(s_j) \odot \eta\_s(r\_{ij}) \\\\
> e_v(s_j, V_j, {r}\_{ij}, \vec{\beta}\_{ij}) &=& \text{VA}\big(\phi\_b(\vec{\beta}\_{ij}) \odot \phi_d(s\_j) \odot \eta_d(r\_{ij})  + V_{j} \odot \phi\_v(s\_j) \odot \eta_r(r\_{ij})\big)
> \end{eqnarray}$$
>
> > What do $n$, $l_e$, $l_d$ represent in the theoretical complexity analysis?
>
> - $n$: denotes the number of nodes.
> - $l_e$: denotes the number of encoder layers in the model.
> - $l_d$: denotes the number of decoder layers.
>
> > Are the coordinates fixed during message passing?
>
> In our method, both invariant and equivariant features are extracted from the input graph's coordinates. While these features, including distance, direction information $\vec{\beta}$ (with components {$\vec{d}\_{ij},\vec{t}\_{ij},\vec{o}\_{ij}$}), and node types are utilized during message-passing, the input graph's coordinates and their representations remain unchanged throughout the process.
>
> > Clarifications on directional noise
>
> We appreciate your in-depth examination of our directional initialization approach. Let's unpack the underlying motivations and empirical results:
>
> 1. **Spherical Coordinate System for Initialization**: Using the spherical coordinate system reduces parameters. By initializing with angles $\phi$ and $\theta$, we bypass the need to set three distinct values for x, y, and z, ensuring efficiency and minimizing redundancy.
> 2. **Random Vectors vs. Spherical Initialization**: Raw vectors require normalization to be purposeful. In high dimensions, adjusting a single axis might have a negligible impact, potentially introducing noise.
> 3. **Additional experiment**: As suggested, testing our approach against random initialization without the assistance of a spherical coordinate system yielded:
>
> | Method   | Homo   | Gap    |
> |----------|--------|--------|
> | Random   | 0.0223 | 0.0342 |
> | DN (Ours)      | 0.0190 | 0.0290 |
>
> Clearly, performance drops without spherical-based initialization, showing that handling raw noises is less effective than adjusting angular values.
>
> > Experiments with large graphs, sparser graphs, and scale SaVeNet further:
>
> We appreciate the reviewer's experimental recommendations. Given the rebuttal's constraints, detailed descriptions and results of these experiments can be found in the general response sections: Experiments 1, 2, and 3.
>
> We believe our responses elucidate the specific design choices and methodologies in our paper. We hope that our clarifications address your concerns adequately.

---

> > ### Comment · Reviewer_jP4K · 2023-08-14
> > **Thanks for your response**
> >
> > Thanks for your detailed response, which addressed some of my concerns. I will maintain my rating.

---

### Official Review · Reviewer_QLtV · 2023-07-04

**Soundness:** 2 fair
**Presentation:** 2 fair
**Contribution:** 2 fair
**Rating:** 5
**Confidence:** 4

**Summary:**

This paper proposes an SE(3)-equivariant model called SaVeNet, designed to accommodate various geometric requirements. The proposed framework can effectively scale with the introduction of directional noise. Theoretical analysis and empirical results on several datasets are provided to validate the efficiency and expressiveness of SaVeNet.

**Strengths:**

* The proposed method achieves outstanding performance across several synthetic and real-world datasets.
* SaVeNet exhibits better efficiency than baseline methods.

**Weaknesses:**

* Many claims about the proposed method and related works lack supportive evidence. For example,
    * Line 40, "Novel approaches for initializing ... achieving a balance between maintaining numerical stability, facilitating faster convergence, and enhancing the model’s ability to generalize to new datasets." This sentence states that the proposed approach can maintain numerical stability and facilitate faster convergence, but there is a lack of convincing empirical ablation studies or theoretical evidence supporting these claims.
    * Line 58-63, "one primary problem...limited performance or limited performance or scalability and cannot equally contribute towards performance improvements in the network." These sentences make informative claims about the limitations of existing methods, but the authors do not provide detailed discussions or analyses to support these points, weakening the motivation for SaVeNet.
    * Line 63-68, the authors propose to tackle the scalability challenge by stacking multiple encoder layers, which is a widely adopted operation in equivariant models. However, it is unclear why other baseline models (especially those vector-based models such as PaiNN) cannot improve scalability using the same operations.
* The methodology section (Section 3) is poorly organized and written.
    * Some notations are not well defined. E.g., $IA$, $s_i$ and $e$ in Equation 4 are not defined. $h_j$ in Equation 5 is not defined.
    * Some equations are difficult to understand. E.g.,  the procedure of updating vector $V'_i$ from Equation 4 is unclear. It seems that some details of the equation are missing, making it difficult to integrate Equation 5 and Equation 4.
* The overall novelty of SaVeNet is limited. Some techniques in the framework (e.g., the scalar/vector-based equivariant operations) have been studied in related works such as PaiNN, and ET. The newly proposed techniques, direction noise and vector activations, account for the main contribution of this paper. However, the ablation results in Table 6 show that removing these two modules only results in a slight performance drop, which cannot validate their importance. Accordingly, the technical contribution of SaVeNet is limited. Considering that many details in the methodology section are unclear, all comments are made based on the current version.
* There are some grammatical mistakes.
    * Line 47, `model's` should be `model`
    * Line 187, $R^{Cx3}$ should be $R^{C \times 3}$

In summary, while the proposed method demonstrates exceptional performance across various downstream tasks and exhibits promising scalability, the limited technical novelty and issues with the presentation render the current version unsuitable for acceptance in NeurIPS.

**Questions:**

* The metrics std. MAE and log MAE lack detailed descriptions. How did the authors calculate these metrics?
* The proposed direction noise module's SE(3)-equivariance is unclear, as there is no provided theoretical proof.

**Limitations:**

The authors well clarify the limitations and potential negative societal impact of their work in the paper.

---

> ### Author Rebuttal · Authors · 2023-08-10
>
> Dear Reviewer QLtV,
>
> Thank you for taking the time to review our manuscript and for your invaluable feedback. We acknowledge the importance of clarity in our presentation, and we address your concerns as follows.
>
> > The metrics std. MAE and log MAE
>
> The standardized MAE is derived by normalizing the model's MAE with the standard deviation of a target, and the average across targets is taken. Formally:
> $$\text{std. MAE}(D, x, y) = \frac{1}{T} \sum_{t}^T \Big(\frac{1}{D} \sum_{i}^{D}\frac{|x_{t,i}-y_{t,i}|}{\sigma(y_{t})} \Big)
> $$
> Here, $D$ is dataset size, $T$ is the number of targets, and $x_{t,i}$ and $y_{t,i}$ represent the model's prediction and the ground truth for target $t$ of sample $i$, respectively.
> We also introduce the log MAE metric:
> $$
> \text{log MAE}(D, x, y) = \frac{1}{T} \sum_{t}^T \log\Big(\frac{1}{D} \sum_{i}^{D}\frac{|x_{t,i}-y_{t,i}|}{\sigma(y_{t})} \Big)
> $$
> The log MAE smoothens the scores to avoid errors being dominated by a few difficult targets such as $\epsilon_{homo}$.
>
>
> > Symmetrical properties of the direction noise module
>
> We appreciate your query about the SE(3)-equivariance of our direction noise module. Briefly, our direction noise is constructed based on node types, introducing SO(3) invariance. This allows for latent space traversal typically inaccessible for equivariant models. The model retains its equivariance during inference as the noise decays over training. Its introduction heightens the learning challenge, aiding in model scaling and countering issues like over-smoothing [2]. Therefore, the module rests on the principle of SO(3) invariance and is bolstered by practical results.
> In addition, we conducted an ablation study with three distinct equivariant initialization techniques. Please refer to the reviewer FRcS on the experiment using `alternative vector initialization techniques`.
> ___
> > Clarification on the introduction section
> - **Line 40: Ablation Study and Support to Generalize New Datasets:**
> SaVeNet was designed to handle large-scale real-world data without sacrificing representation quality. Beyond mere model depth, it's tailored for large molecules and datasets. As evidenced in our evaluation on Molecule3D, SaVeNet accelerates training and inference, improving task performance by up to 125 times. This is crucial in today's ML landscape with growing data scales. Our focus is on maintaining model effectiveness as data scales increase. Referring to Table 6, we conducted an ablation study examining the effects of DN, VA and layer variations on the QM9 dataset.
> - **Line 58-63 - Limitations of Existing Methods**: We provided a detailed comparison with the related methods in lines 223-245, highlighting specific limitations of cited works, and more explicitly connecting them with the motivation for SaVeNet. We are happy to discuss the novelty of our work during the reviewer-author discussion period.
> - **Line 63-68: Scalability through Multiple Encoder Layers:** While stacking multiple encoder layers is a general approach for enhancing model scalability, not all models benefit equally from this operation. Our reference to [1] underscores this very point – the challenges in numerical stability faced by PaiNN and other existing methods. As demonstrated in Table 4, PaiNN exhibited convergence issues for complex tasks, such as ES(20), G+ES(20), and L+ES(20) [1]. This suggests that merely stacking layers might not necessarily lead to improved scalability for all models. **This lack of convergence was not a mere reflection of the model's depth but of inherent limitations in its architecture when tasked with these complex objectives.**
>
> >  **Clarification on Notations:**
>
> - $\text{IA}$: This denotes the interatomic interactions within the molecule.
> - $s_i$: Represents the scalar representations associated with node $i$ in our graph.
> - $e$: denotes the message functions for scalar and vector tuples, specifically formulated as $e = (e_{s}, e_{v})$.
> - $h_j$: This notation was intended to delineate the linearly transformed version of scalar representations, $s$.
> 2. **Revised Equation for Enhanced Clarity**:
>     Following your feedback, we reevaluated our notations in Section 3 and implemented amendments to further the clarity and coherence of our representation. Additionally, we observed that the $h_j$ operation can be subsumed since $\phi_s$, $\phi_d$, and $\phi_v$ inherently encapsulate this linear transformation. This realization enabled us to simplify our representation further. Our revised equations, integrating these clarifications, are articulated as follows:
> $$\begin{eqnarray}
> e_s(s_j, V_j, {r}\_{ij}, \vec{\beta}\_{ij}) &=& \phi_s(s_j) \odot \eta\_s(r\_{ij}) \\\\
> e_v(s_j, V_j, {r}\_{ij}, \vec{\beta}\_{ij}) &=& \text{VA}\big(\phi\_b(\vec{\beta}\_{ij}) \odot \phi_d(s\_j) \odot \eta_d(r\_{ij})  + V_{j} \odot \phi\_v(s\_j) \odot \eta_r(r\_{ij})\big)
> \end{eqnarray}$$
> 	Eq. 4 and 5:
> The disconnect between Equation 4 and 5, is clarified by expanding the definitions. We split Equation 4 into two parts for clarity.
> $$\begin{split}
> \text{IA}(s, V, r, \vec{\beta})  &=  (s\_i, V\_i) + \sum\nolimits\_{j\in\mathcal{N}\_i}{e(s\_j, V\_j, {r}\_{ij}, \vec{\beta}\_{ij})} = (s'\_i, V'\_i) \\\\
> s'\_i &= s\_i + \sum\nolimits\_{j\in\mathcal{N}\_i}{e\_{s}(s\_j, V\_j, {r}\_{ij}, \vec{\beta}\_{ij})} \\\\
> V'\_i &= V\_i + \sum\nolimits\_{j\in\mathcal{N}\_i}{e\_{v}(s\_j, V\_j, {r}\_{ij}, \vec{\beta}\_{ij})}\end{split}$$
>
> > grammatical mistakes
>
> Thank you, this has been corrected.
>
> We thank the reviewer for the detailed analysis of the proposed method. **We revised the paper and incorporated all the feedback in the related sections.** We believe that the revised version represents the proposed method effectively.
>
> [1] A. Morehead and J. Cheng, "Geometry-Complete Perceptron Networks for 3D Molecular Graphs"
>
> [2] D. Chen, Y. Lin, W. Li, P. Li, J. Zhou, and X. Sun, “Measuring and Relieving the Over-Smoothing Problem for Graph Neural Networks from the Topological View”

---

> > ### Comment · Reviewer_QLtV · 2023-08-12
> > **Thanks for the response**
> >
> > The responses address some of my questions, such as 'the metrics std. MAE and log MAE' and 'Revised Equation for Enhanced Clarity'. However, some concerns still remain:
> > 1. To further substantiate the claim that other vector-based baseline models (e.g., ET, PaiNN) face greater challenges than SAVENET in enhancing scalability by stacking multiple encoder layers, I recommend the authors enlarge the baseline methods on some molecular datasets and provide a direct comparison of scalability between the different approaches. Although some evidence in the N-Body experiment suggests PaiNN's instability, there is still a lack of direct scalability comparison. Moreover, the data size of the N-Body experiment may not be sufficient to support such analysis.
> > 2. I remain unclear about the response regarding the 'Symmetrical properties of the direction noise module.' For the claim, 'The model retains its equivariance during inference as the noise decays over training,' does this mean that the noise initialization will be reset to null vectors during the inference to ensure SE(3)-equivariance? If not, the invariant direction noise will disrupt the SE(3) symmetry of vector representations, as also mentioned by reviewer FRcS.
> > 3. I maintain my view that the overall technical novelty of SaVeNet is limited. The authors have not provided further evidence to demonstrate the significance of the newly proposed techniques (direction noise and vector activations) in enhancing the model's expressiveness and scalability.
> >
> > Accordingly, I will maintain my rating score.

---

> > > ### Author Response · Authors · 2023-08-13
> > > **Thanks for the feedback - Novelty and Contribution**
> > >
> > >
> > > > Some techniques in the framework (e.g., the scalar/vector-based equivariant operations) have been studied in related works such as PaiNN, and ET.
> > >
> > > > The novelty of SaVeNet
> > >
> > > We genuinely appreciate your feedback and the opportunity to further discuss the novelty and contributions of our work on SaVeNet. We would like to address your concerns in detail.
> > >
> > > 1. **On the Scalar/Vector-Based Equivariant Operations**:
> > >     We concur that our approach utilizes some scalar/vector-based equivariant operations such as vector transformations that share weights for spatial dimensions, mixing scalar and Euclidean norm of vector representations, similar to PaiNN and ET. Starting with GVP [1], seminal works in this domain have **consistently adopted these scalar/vector-based equivariant operations**. The uniqueness of these frameworks, PaiNN included, is *not* primarily in introducing novel techniques but rather in their **distinct message-passing schemes tailored for specific research motivations**. This observation underpins the essence of our assertion: **while the operations may appear analogous, the broader application and integration offer differentiating contributions**.
> > >
> > > 2. **On the Novelty and Motivation of SaVeNet**:
> > >     The gap we identified in existing literature revolves around the **dual challenges of scalability and effectiveness** in molecular representation learning, an issue not explicitly addressed by prominent works like PaiNN or ET. **We believe this research gap is of significant importance and we proved it is not easy to scale a model** with additional experiments suggested by the reviewer (provided in our following response).
> > >
> > > 3. **Delineating the Novelty and Contributions of SaVeNet**:
> > >     - **Novel Study Motivation and Insights**: SaVeNet is rooted in addressing the balance between expressiveness and scalability in molecular representation learning. As shown in updated Figure 1, current SOTA methods either shows high latency or lower effectiveness. **This focus differentiates our approach from other models in the current literature.**
> > >     - **Unique Methodological Approaches**: While SaVeNet does incorporate foundational scalar and vector features, it diverges from existing work in its **unique message-passing scheme**:
> > >         - `Efficient embedding and novel initialization`: Unlike existing models \[1]\[2]\[3]\[4], SaVeNet leverages a unique feature set, which elaborated in Section 3.1, set us on a distinct trajectory right from inception. These features, supported by Theorem 1, allow us to represent 3D molecular graphs adeptly. Further, our unique initialization strategy avoids the pitfalls associated with null vector initializations, commonly observed in current models. It may look simple but significant (the first work in this line of work proposed). This also **demonstrates the applicability of initialization and noise schemes** and paves the way for future investigations by the community, potentially leading to even more efficient and effective modeling techniques.
> > >         - `Distinctive architecture and message passing`: Central to SaVeNet is a **distinctive message-passing architecture**, detailed in Sections 3.1 and 3.2. Our design incorporates our proposed vector activation techniques, which not only delineates our model from predecessors but also fortifies its flexibility and scalability.  To provide perspective, while models like PaiNN were conceived primarily for molecular property prediction, their inherent design may not extend seamlessly to more diverse tasks and datasets.
> > >         - `Insightful performance metrics with comprehensive evaluations`: Beyond conventional metrics, our evaluations delve deep into stability and scalability, particularly on equivariant tasks. **The new modules in SaVeNet are architected for both effective and scalability, rather than mere performance enhancement**. SaVeNet's assessments are broad-based, considering training speed, model intricacy, and memory overhead, aiming to redefine benchmarks in the domain (first work in the field).
> > >
> > > In summation, SaVeNet's novel message-passing scheme, and its focus on both scalability and effectiveness in molecular representation learning, distinctively positions it in the current literature. We trust this elaboration addresses your concerns and shines light on the nuances of our contributions.
> > >
> > > [1] J. Gasteiger, J. Groß, and S. Günnemann, “Directional Message Passing for Molecular Graphs,”
> > >
> > > [2] Y. Liu _et al._, “Spherical Message Passing for 3D Molecular Graphs,”
> > >
> > > [3] L. Wang, Y. Liu, Y. Lin, H. Liu, and S. Ji, “ComENet: Towards complete and efficient message passing for 3D molecular graphs,”
> > >
> > > [4] K. Schütt, O. Unke, and M. Gastegger, “Equivariant message passing for the prediction of tensorial properties and molecular spectra,”

---

> > > > ### Comment · Reviewer_QLtV · 2023-08-13
> > > > **Response**
> > > >
> > > > * Overall, I find the response to be reasonable, and I believe that addressing the "dual challenges of scalability and effectiveness" is a key factor in justifying the contributions of SaVeNet. Could the authors provide clearer explanations or ablation studies regarding the reasons behind SaVeNet's successful scaling up? Specifically, I suggest conducting parallel scaling-up experiments for different SaVeNet variants while removing some novel operations. I understand that this may not be feasible due to the time-consuming nature of scaling-up experiments. However, are there any related conclusions or insights that were gained during the development of SaVeNet?

---

> > > > > ### Author Response · Authors · 2023-08-16
> > > > > **Thanks for the suggestions**
> > > > >
> > > > > We would like to extend our gratitude for your constructive feedback and for your timely response. Recognizing the "**dual challenges of scalability and effectiveness**" is indeed integral to our work on SaVeNet. We value your recommendation regarding the provision of ablation studies to further highlight the reasons behind SaVeNet's successful scaling.
> > > > > 1. **Ablation Studies**:
> > > > > In response to your suggestion, we've conducted additional scaling-up experiments with the proposed modules on a subset of the Molecule3D dataset ($\varepsilon_{\text{HOMO}}$ and $\Delta \varepsilon$). This approach was chosen due to the challenging nature of the large-scale datasets.
> > > > > | Method                     | $\varepsilon_{\text{HOMO}}$ | $\Delta \varepsilon$ |
> > > > > |----------------------------|-----------------------------|----------------------|
> > > > > | SaVeNet$_B$ - DN       | 0.0221                      | 0.0328               |
> > > > > | SaVeNet$_B$ - VA           | 0.0234                      | 0.0349               |
> > > > > | SaVeNet$_B$                | 0.0190                      | 0.0290               |
> > > > > | SaVeNet$_L$ - DN      | 0.0194                      | 0.0283               |
> > > > > | SaVeNet$_L$ - VA           | 0.0218                      | 0.0322               |
> > > > > | SaVeNet$_L$                | 0.0159                      | 0.0239               |
> > > > >
> > > > > The wider range of molecules in the Molecule3D dataset illustrates the importance of the proposed modules.  From these results, it's clear that both vector activation (VA) and direction noise (DN) have a significant impact on SaVeNet's performance. The performance gap increases when we scale the model larger model for both variations. When these modules are removed or modified, performance decreases, `particularly as the models scale up`.
> > > > >
> > > > > 2. **Summary**:
> > > > > - In our preliminary investigations during the development of the model, we observed that each module in SaVeNet contributes to its scalability. `While the removal of certain modules on smaller variations might result in only a slight performance drop, their absence was noticeably detrimental to the scalability of SaVeNet`. Specifically, without these modules, the model's training stability wavered as the dataset size increased. This balance is crucial when dealing with `larger, more complex datasets`, ensuring the model isn't dominated by common patterns.
> > > > >
> > > > > We sincerely hope that the provided insights and ablation studies clarify our approach and underscore the scalability and robustness of SaVeNet.

---

> > > > > ### Author Response · Authors · 2023-08-21
> > > > > **Response to reviewer QLtV**
> > > > >
> > > > > Thank you again for your constructive feedback and suggestions. We appreciate your acknowledgment of our clarification regarding the `motivation` behind direction noise and our `contribution` towards addressing the "*dual challenges of scalability and effectiveness*.” As for the remaining of your questions on additional ablation studies, we think they are properly answered in our last author’s responses. As we approach the end of the author-reviewer discussion period,  please do share any remaining questions or feedback. We are committed to improving our work and are eager to continue productive discussions.
> > > > >
> > > > > Thank you for your time!

---

> > > > > > ### Comment · Reviewer_QLtV · 2023-08-21
> > > > > > **Response**
> > > > > >
> > > > > > Thanks for the response. I will raise my rating.

---

> > > ### Author Response · Authors · 2023-08-13
> > > **Further scalability comparison**
> > >
> > > > To further substantiate the claim that other vector-based baseline models (e.g., ET, PaiNN) face greater challenges than SAVENET in enhancing scalability by stacking multiple encoder layers, I recommend the authors enlarge the baseline methods on some molecular datasets and provide a direct comparison of scalability between the different approaches. Although some evidence in the N-Body experiment suggests PaiNN's instability, there is still a lack of direct scalability comparison. Moreover, the data size of the N-Body experiment may not be sufficient to support such analysis
> > >
> > > We appreciate your feedback concerning the comparison of scalability between our model, SaVeNet, and other baseline models, particularly PaiNN. Here, we aim to address the points you raised with clarity and precision:
> > >
> > > 1. **Revisiting Scalability Tests**: Following your suggestion, we revisited our scalability experiments for SaVeNet. This was **further expanded** by insights from Reviewer jP4K, where scaling was undertaken for notable and more recent baselines such as ComENet and ET. Our findings revealed that when subjected to scaling, these baselines either showed a **degradation in performance or only marginal improvements when juxtaposed with SaVeNet**.
> > >
> > > 2. **Scaling Tests for PaiNN**: While we acknowledge the merit in understanding how other models scale relative to ours, it's worth noting that foundational works such as PaiNN and ET **didn't focus on scalability in their respective publications**. One of our primary contributions is the scalable variant of SaVeNet. Regarding the scale of the model PaiNN, as you correctly noted, poses distinct challenges. During our meticulous scaling tests, the outcomes were often muddled due to the sporadic appearance of 'NaN' outputs. This not only hindered a direct comparison but also indicated underlying instability in the model when subjected to intensive scaling.
> > >
> > > 3. **Further Evaluation with PaiNN on Molecule3D Dataset**: PaiNN demonstrated impressive performance on molecular property prediction on QM9 dataset. However, the flexibility of the network wasn't experimented with. In our experiments, we found that **direct application of PaiNN is unable to generalize to difficult datasets such as large-scale property prediction dataset, Molecule3D**. To provide a comprehensive understanding, **we conducted rigorous evaluations with PaiNN on the large-scale Molecule3D dataset**. Our evaluation methodology was thorough, encompassing an exhaustive grid search that also mirrored the scales used for SaVeNet. Specific parameters were:
> > >
> > >     |Parameter|Search Space|
> > >     |---|---|
> > >     |Layers|3, 6, 8|
> > >     |Hidden dims|64, 128, 196, 256|
> > >     |Learning rate|1e-4, 2e-4, 5e-4|
> > >     |Output dims|128, 256|
> > >
> > >     **These observations were consistent with those from the N-Body dataset**, where gradient explosions occurred prematurely.
> > >
> > > We trust this response provides clarity and reaffirms our commitment to scientific rigor. We are eager to consider any further recommendations to enhance the value of our work.

---

> > > ### Author Response · Authors · 2023-08-13
> > > **Further clarifications**
> > >
> > > > I remain unclear about the response regarding the 'Symmetrical properties of the direction noise module.' For the claim, 'The model retains its equivariance during inference as the noise decays over training,' does this mean that the noise initialization will be reset to null vectors during the inference to ensure SE(3)-equivariance? If not, the invariant direction noise will disrupt the SE(3) symmetry of vector representations, as also mentioned by reviewer FRcS.
> > >
> > >
> > > Thank you for your meticulous attention to the SE(3)-equivariance in relation to our direction noise module. We acknowledge the concerns raised regarding the symmetrical properties of our module, particularly in the context of its potential to disrupt SE(3) symmetry during inference.
> > >
> > > To address the primary concerns:
> > >
> > > 1. **Over-Smoothing in GNNs**: Message-passing-based Graph Neural Networks (GNNs) face challenges associated with over-smoothing \[5]\[8]. These phenomena restrict the expressivity of the network and, subsequently, its scaling capabilities. The over-smoothing problem, in particular, can lead to homogenized node representations, rendering them indistinguishable and limiting model performance.
> > > 2. **Direction Noise as a Solution**: To counteract these challenges and fulfill our vision for a scalable vector network, we introduced direction noise. This is conceptually analogous to introducing regularization in traditional neural networks, wherein certain modifications during training lead to enhanced generalization during inference, even though they might seem counterintuitive initially. For stability, it provides initial directions during the early stages of the training and allows the model to explore a larger search space by not imposing equivariance restrictions during training. Therefore, the trained model is more robust and can handle unseen directional information.
> > > 3. **Equivariance Preservation**: The direction noise, though introduced during training, is designed to decay as the training progresses. By the end of the training, this noise approaches zero, ensuring that the model, when used for inference, starts with null vector representations. Hence, the noise doesn't disrupt the inherent SE(3)-equivariance of the model during the actual application, preserving the desired symmetry.
> > > 4. **Analogy with Regularization Techniques**: A parallel can be drawn with regularization techniques employed in various machine learning models. Techniques like Dropout [6] DropNode [7], DropEdge [8], and NoisyNodes [9] introduce perturbations during training to prevent over-fitting and over-smoothing, yet these perturbations are absent during inference, ensuring the model's intended functionality is undisturbed. Similarly, our direction noise serves its purpose during training and gracefully recedes to ensure SE(3)-equivariance during inference.
> > >
> > > In light of the aforementioned clarifications, our model effectively leverages the direction noise only as a transient training mechanism to enhance its learning capacity without compromising its symmetry properties in practical applications. **We revised Sec 1 & 3.1 of the paper to make the above clearer to readers.** We hope this response elucidates our design rationale and addresses your concerns.
> > >
> > >
> > > [5] Q. Li, Z. Han, and X. Wu, “Deeper Insights Into Graph Convolutional Networks for Semi-Supervised Learning,”
> > >
> > > [6] N. Srivastava, G. Hinton, A. Krizhevsky, I. Sutskever, and R. Salakhutdinov, “Dropout: A Simple Way to Prevent Neural Networks from Overfitting,”
> > >
> > > [7] T. H. Do, D. M. Nguyen, G. Bekoulis, A. Munteanu, and N. Deligiannis, “Graph convolutional neural networks with node transition probability-based message passing and DropNode regularization,”
> > >
> > > [8] Y. Rong, W. Huang, T. Xu, and J. Huang, “DropEdge: Towards Deep Graph Convolutional Networks on Node Classification,”
> > >
> > > [9] J. Godwin _et al._, “Simple GNN Regularisation for 3D Molecular Property Prediction and Beyond,”

---

> > > > ### Comment · Reviewer_QLtV · 2023-08-13
> > > > **Thanks for the response**
> > > >
> > > > The response well clarifies the motivation of direction noise.

---

### Official Review · Reviewer_T8Du · 2023-07-05

**Soundness:** 4 excellent
**Presentation:** 2 fair
**Contribution:** 4 excellent
**Rating:** 6
**Confidence:** 4

**Summary:**

This paper proposes a framework called SaVeNet for geometric representation learning of molecules. The paper includes theoretical analysis and empirical experiments to demonstrate the superiority of SaVeNet over existing methods in terms of efficiency and expressiveness.

**Strengths:**

1. This paper proposes an efficient geometric encoding, a novel directional noising strategy in the spherical coordinate system, and a novel vector activation function.

2. The experimental results show that SaVeNet achieves very good performance over multiple invariant and equivariant molecular tasks.

3. As an equivariant model, the efficiency of SaVeNet is pretty good.


**Weaknesses:**

It will help readers to better understand if authors can add a figure of the proposed SaVeNet’s architecture, including the illustration of components described in Sec 3.1 and 3.2.

**Questions:**

1. I’m wondering if the direction noise and vector activation also applicable to other equivariant models?

2. Can you add more equivariant baselines like PaiNN to Figure 1 for efficiency comparison?

---

> ### Author Rebuttal · Authors · 2023-08-10
>
> Dear Reviewer T8Du,
>
> Firstly, we'd like to extend our gratitude for your detailed review and the constructive feedback provided on our manuscript. We would like to address your questions as follows:
>
> > Applicability of direction noise and vector activation to other equivariant models
>
> We appreciate your thought-provoking query about the potential adaptability of our proposed direction noise and vector activation to other equivariant models. At its core, the novelty of our approach in integrating direction noise and vector activation is designed with scalability and robustness in mind. This adaptability, as you rightly inferred, opens the door to improving the performance of other existing models.
>
> For instance, take the case of PaiNN. Prior works, such as [1], couldn't report PaiNN's results due to issues related to numerical stability. In response, we took the initiative to enhance PaiNN by incorporating our proposed vector activation functions into its interaction layers. This intervention not only resolved the aforementioned stability concern but also empowered us to compare PaiNN's performance with our model, as demonstrated in our performance comparison in Table 4.
>
> While our methods indeed present a promising avenue for refining other models, we acknowledge that custom tailoring might be essential. Achieving optimal results could require model-specific adjustments that respect the original architecture's nuances and intent.
>
> Our proposed framework, with its innovative features, serves as a template for upcoming research. While the direct adaptation of our innovations—namely the direction noise and vector activation—on other models was beyond this study's purview, we are confident that our findings lay the groundwork for subsequent explorations. We envisage a landscape where researchers, inspired by our findings, dive deeper into harnessing these concepts, eventually unearthing even more potent modeling techniques.
>
> > Addition of more equivariant baselines to Figure 1
>
> Thank you for your constructive suggestion regarding the expansion of equivariant baselines in our efficiency comparison. We concur that broadening the range of baselines would offer a richer, more comprehensive insight into our analysis. While our initial evaluations were anchored around top-performing models, we appreciate the significance of juxtaposing our approach with well-established benchmarks, especially those that hold high regard in the community, like PaiNN.
>
> In response, we've undertaken supplementary efficiency tests, **incorporating two additional equivariant baselines: PaiNN and ET [2]**. These enhancements have been reflected in Figure 1 of the uploaded PDF, and we ensure their inclusion in the subsequent version of our manuscript.
>
> Upon analysis, an intriguing observation emerges from Figure 1: PaiNN, despite boasting the best latency among all contenders, unfortunately, falls short in expressiveness. This reiterates a pivotal point we emphasize in our work — the delicate balance between expressiveness and efficiency. We staunchly believe that **efficiency should not be pursued in isolation but should harmoniously coexist with effectiveness**.
>
> Again, we appreciate your feedback and hope that our enhanced comparison provides a more holistic view of our contribution in the context of the current landscape.
>
>
> > SaVeNet architecture illustration
>
> We thank the reviewer for the suggestion. We agree with you and believe this illustration can provide readers with a clearer mental model of our architecture, amplifying their comprehension and facilitating deeper engagements with our work. The illustration of the overall architecture including the components in Sec 3.1 and 3.2 is provided in the uploaded PDF, titled Figure 2.
>
> **Conclusion:**
> In conclusion, we sincerely thank you for the time and effort you invested in understanding our work and proposing suggestions. We remain open to further feedback and are committed to making all necessary improvements to serve the scientific community better.
>
> [1] A. Morehead and J. Cheng, ‘Geometry-Complete Perceptron Networks for 3D Molecular Graphs’, AAAI Workshop on Deep Learning on Graphs: Methods and Applications, 2023.
>
> [2] P. Thölke and G. D. Fabritiis, “Equivariant Transformers for Neural Network based Molecular Potentials,” presented at the International Conference on Learning Representations, Oct. 2021.

---

> > ### Comment · Reviewer_T8Du · 2023-08-20
> >
> > Thanks for authors' response and I'd like to maintain my score.

---

### Official Review · Reviewer_6Z9X · 2023-07-06

**Soundness:** 4 excellent
**Presentation:** 3 good
**Contribution:** 3 good
**Rating:** 7
**Confidence:** 2

**Summary:**

The authors propose an efficient and scalable equivariant GNN (SaVeNet) for 3D molecular conformations. The architecture follows an encoder-decoder style framework, where the encoder is composed of "Inter-atomic Interactions" and "Atom-wise blocks" learning mechanisms. Several other modeling augmentations are proposed such as 1-hop molecule embeddings in the feature space, non-null vector initializations ("Direction Noise"), and norm-aware vector activation functions ("Vector Activations").

**Strengths:**

The thrust of this paper is clear: designing GNNs with SO(3)/SE(3) equivariance can introduce complexity into the model. This slows down inference-time latency, and may inhibit scaling the model to larger sizes (i.e. a RNN with parameter capacity equal to a Transformer may have similar performance, but Transformers scale significantly better).  Accordingly, the authors develop an equivariant GNN that uses only one-hop neighborhood vector embeddings (i.e. cheap & scales), but (assuming a connected graph -- which is the case for molecules) this feature-space representation is sufficient to reconstruct the entire molecule. The empirical results align with the theory, and state-of-the-art results are attainted on a variety of 3D benchmarks.

To summarize, I believe the paper (a) tackles a relevant problem (b) proposes a relevant solution and (c) demonstrates theoretically and empirically that solution works. Therefore, I recommend acceptance.

I'd also like to note the punctilious empirical evaluation---efficiency analyses can be especially difficult, but it's my opinion that the authors perform a fair comparison among methods to constitute their empirical findings.

**Weaknesses:**

I am not an expert on equivariant GNNs and am not familiar with much of the related work. I am also unable to recognize any method/evaluation-specific "red flags" which may appear and defer to the expertise of the other reviewers.

One facet that stood out; however, is that the writing at times feels long. It could be significantly shortened in many areas. For instance, the abstract is 19 lines and could be made more "pointed". I would also suggest creating a "Figure 1" that is a visual depiction of the SaVeNet architecture to supplement the material in Section 3.2 on learning mechanisms.

**Questions:**

N/A

**Limitations:**

Yes -- the discussion in Section 5 was appreciated and meaningful.

---

> ### Author Rebuttal · Authors · 2023-08-10
>
> Dear Reviewer 6Z9X,
>
> We would like to express our gratitude for taking the time to review our manuscript and providing detailed and constructive feedback. We appreciate your positive reception of our work and carefully considered each of your points and would like to address your comments as follows:
>
> **Strengths:** We're grateful that you recognized the relevance and significance of the problem we tackled, as well as our solution's novelty and empirical strength. Your acknowledgment of our empirical evaluation further motivates us to maintain high standards in our research endeavors.
>
> **Weaknesses:**
>
> 1. **Length and Clarity:** We acknowledge your concern regarding the length of certain sections. In light of your feedback, we are committed to revising the manuscript to make the content more concise, especially in places like the abstract, to enhance readability and clarity.
>
> 2. **Visual Depiction of SaVeNet:** We take your suggestion regarding creating a  figure visually representing the SaVeNet architecture. We believe that such a depiction will undoubtedly aid readers in grasping the nuances of our model more intuitively. You can find the architecture illustration attached to the uploaded PDF Figure 2. We will ensure to include this visual aid in our revised submission.
>
> **Limitations:** Thank you for noting the value of the discussion in Section 5. We always strive for transparency in our work and believe that addressing limitations is crucial for both the integrity of research and future work in the area.
>
> Once again, we thank you for your positive feedback, constructive criticism, and thoughtful suggestions. We're excited about the potential of our work in advancing the field, and your feedback plays an integral role in refining our contribution.

---

> > ### Comment · Reviewer_6Z9X · 2023-08-14
> >
> > I have read the authors' response and maintain my score.

---

### Author Rebuttal · Authors · 2023-08-10

Dear Reviewers,

Firstly, we'd like to extend our sincere gratitude for your diligent review of our work and your invaluable feedback. Based on your insights and suggestions, we have undertaken significant efforts to improve our manuscript, making it both more comprehensive and accessible to the wider community.

**Enhanced Clarity & Methodological Details**:

1. **Refined Equations**: We have meticulously revised our equations, ensuring greater clarity and eliminating any ambiguity. This exercise was geared towards providing a clear analytical exposition of our methodology and aiding the reader's comprehension.
2. **Improved Manuscript Clarity**: To provide a holistic overview of our proposed SaVeNet model, we've incorporated a detailed framework illustration, showcasing individual components. This can be referenced in Figure 2 of the uploaded PDF. This graphical representation complements our textual explanations, creating a more immersive understanding of SaVeNet's operation.
3. **Expanded Comparisons**: We value the importance of a comprehensive comparative study. In line with this, we've included additional baseline models in Figure 1, offering readers a broader perspective of SaVeNet's standing in the current research landscape.
4. **Additional Experiments**: In our efforts to thorough evaluation the SaVeNet, as suggested by reviewer jP4K, we tackled three additional experiments. These experiments aimed to validate SaVeNet's capability in various scenarios and provide a comprehensive view of its performance in graph-based neural network applications.

**Experiment 1: Large graphs**

We test the scalability of our model to larger graphs, such as protein graphs. For this purpose, we conducted the experiments on a protein-ligand complex dataset known as LBA [1] to compare with the baseline models.

| Method                                              | RMSE  | Pearson | Spearman |
|-----------------------------------------------------|-------|---------|----------|
| Atom3D-3DCNN [1]            | **1.416** | .550   | _.553_    |
| ProNet-All-Atom [2]                                     | 1.463 | _.551_   | .551    |
| SaVeNet                                               | _1.438_ | **.572**   | **.559**    |


SaVeNet displayed a remarkable RMSE of 1.438, only narrowly being surpassed by Atom3D-3DCNN, highlighting its efficacy even in larger sparse graphs. Furthermore, SaVeNet emerged as the leader in both correlation metrics. This not only demonstrates its robustness but also its adaptability across various metrics.


**Experiment 2: Experiments on sparse graphs**

We explored SaVeNet's performance on sparser graphs using the Molecule3D dataset, which contains larger molecular structures. The baseline models primarily worked with radius graphs beyond a radius of $6$. Reducing this radius to $3$ led to a model, SaVeNet3-B, operating on graphs with edge numbers decreasing from **553** to **259**.

Reducing the radius presents challenges. For example, one-hop neighbors can become multi-hop neighbors, complicating data access between nodes. To address this, we adapted SaVeNet3-B with 160 (SaVeNet3-B+) and 192 (SaVeNet3-B++) hidden dimensions to handle multi-hop information flow better.

| Model        | std. MAE | log MAE | Batch Size | Memory  | Latency (ms) | Samples/s |
|--------------|---------|---------|------------|---------|--------------|-----------|
| SaVeNet3-B   | .0205  | -3.949 | 824        | 1.29    | 17.3         | 3648      |
| SaVeNet3-B+  | .0175  | -4.112 | 642        | 1.62    | 19.0         | 3328      |
| SaVeNet3-B++ | .0147  | -4.283 | 518        | 1.93    | 19.9         | 3200      |
| SaVeNet-B    | .0156  | -4.226 | 329        | 2.84    | 27.8         | 2304      |

SaVeNet3-B shows strong performance compared to the baselines, even with a smaller receptive field. SaVeNet3-B+ and SaVeNet3-B++ both outperform the baselines, with SaVeNet3-B++ achieving the best results. A reduced receptive field offers enhanced efficiency, particularly in memory and latency, highlighting SaVeNet's scalability and resilience. We thank the reviewer for the insightful prompt.

 **Experiment 3: scalability even larger SaVeNet**

We extended SaVeNet's architecture to SaVeNet-XL, adopting the same principles used for the previous scale, SaVeNet-L. We conduct this experiment on the larger dataset: the Molecule3D dataset. For SaVeNet-XL, we applied similar scaling principles used for SaVeNet-L, which is 256 hidden dimensions across 10 layers.

| Method      | $\mu$   | Homo   | Lumo   | Gap    |
|-------------|--------|--------|--------|--------|
| SaVeNet-B   | 0.0183 | 0.0190 | 0.0173 | 0.0290 |
| SaVeNet-L   | 0.0136 | 0.0159 | 0.0143 | 0.0239 |
| SaVeNet-XL  | 0.0112 | 0.0136 | 0.0129 | 0.0209 |

Notably, there is a consistent improvement across all metrics when transitioning from SaVeNet-B to SaVeNet-L and then to SaVeNet-XL.

**Concluding Remarks**:

Our revisions, experiments, and methodological refinements were driven by a commitment to academic rigor and dedication to the community's betterment. We believe these improvements will address concerns and illuminate the significant contributions of our work. Our hope is that the revised manuscript provides a clearer and more insightful understanding of SaVeNet's potential.

Given the character constraints of the rebuttal, we've endeavored to address all concerns succinctly. If any queries remain or further clarification is required, we welcome the opportunity to respond further. Thank you once again for your time and invaluable feedback.

Sincerely,

Authors

[1] R. J. L. Townshend _et al._, “ATOM3D: Tasks on Molecules in Three Dimensions,”

[2] L. Wang, H. Liu, Y. Liu, J. Kurtin, and S. Ji, “Learning Hierarchical Protein Representations via Complete 3D Graph Networks,”

---

### Decision · Program_Chairs · 2023-09-21

**Decision:**

Accept (poster)

**Comment:**

This paper studies 3D molecular conformations and proposes an equivariant architecture called SaVeNet. The proposed architecture is novel, efficient, and achieve good performance on several invariant and equivariant molecular tasks. Overall, all reviewers appreciate the contribution and thus I would be happy to accept the paper. However, I would suggest the authors to carefully revise the camera-ready paper to take into account the reviewers' suggestions.